# EPHX1 and ERCC2 polymorphisms are associated with cisplatin-induced nephrotoxicity and prognosis in Thai cancer patients

Saad Ahmed[1], Jakris Eu-ahsunthornwattana [2,3], Thanaporn Thamrongjirapat[4], Aruchalean Taweewongsounton[5], Yutthana Rittavee[1,6], Nintita Sripaiboonkit Thokanit [7], Montien Ngodngamthaweesuk[8], Pitichote Hiranyatheb[9], Thanyanan Reungwetwattana[4], Nuttapong Ngamphaiboon[4], Natini Jinawath [1,10,6]*

1 Program in Translational Medicine, Faculty of Medicine Ramathibodi Hospital, Mahidol University, Bangkok, Thailand, 2 Department of Community Medicine, Faculty of Medicine Ramathibodi Hospital, Mahidol University, Bangkok, Thailand, 3 Division of Medical Genetics and Molecular Medicine, Department of Internal Medicine, Faculty of Medicine Ramathibodi Hospital, Mahidol University, Bangkok, Thailand, 4 Division of Medical Oncology, Department of Medicine, Faculty of Medicine Ramathibodi Hospital, Mahidol University, Bangkok, Thailand, 5 Research Center, Faculty of Medicine Ramathibodi Hospital, Mahidol University, Bangkok, Thailand, 6 Chakri Naruebodindra Medical Institute, Faculty of Medicine Ramathibodi Hospital, Mahidol University, Bang Phli, Samut Prakarn, Thailand, 7 Ramathibodi Comprehensive Cancer Center, Faculty of Medicine Ramathibodi Hospital, Mahidol University, Bangkok, Thailand, 8 Division of Cardiovascular Thoracic Surgery, Department of Surgery, Faculty of Medicine Ramathibodi Hospital, Mahidol University, Bangkok, Thailand, 9 Division of General Surgery, Department of Surgery, Faculty of Medicine, Ramathibodi Hospital, Mahidol University, Bangkok, Thailand, 10 Integrative Computational BioScience Center (ICBS), Mahidol University, Nakhon Pathom, Thailand

* natini.jin@mahidol.ac.th, jnatini@hotmail.com

## Abstract

Cisplatin is a widely used chemotherapeutic drug for various cancers. One of the common adverse effects of cisplatin is nephrotoxicity including acute kidney injury (AKI) and acute kidney disease (AKD). Single Nucleotide Polymorphisms (SNPs) can be used to identify cancer patients who are susceptible to developing cisplatin-induced nephrotoxicity (CIN). In this study, we validated the association between 6 SNPs in the drug metabolizing enzyme genes, *SLC22A2* (rs316019) & *EPHX1* (rs1051740), and the DNA repair genes, *ERCC1* (rs11615 & rs3212986) & *ERCC2* (rs13181 & rs1799793), and CIN in the 169 Thai patients with head and neck, lung, or esophageal cancer. Effect of these SNPs on cumulative incidence of AKD, progression-free survival (PFS), and overall survival (OS) was also assessed. *EPHX1* rs1051740 TC genotype was significantly associated with AKD in co-dominant [OR 2.894, 95% CI 1.091–7.680; $P = 0.033$] and over-dominant [OR 2.793, 95% CI 1.333–5.851; $P = 0.006$] models, and with an increased cumulative incidence of AKD ($P = 0.021$). Additionally, *ERCC2* rs13181 and rs1799793 were significantly associated with OS ($P = 0.002$ and 0.004). Our results reveal an association between *EPHX1* rs1051740 and AKD, and confirms the previously reported associations between *ERCC2* SNPs and OS. These findings may help in predicting CIN in Thai cancer patients.

**Data availability statement:** All relevant data are within the manuscript and its supporting information files. We have removed all identifying information. The genotyping information and clinical metadata required to replicate the reported study findings can be found in S9 Data file.

**Funding:** This research project is supported by Mahidol University, and the National Science Research and Innovation (NSRF) via the Program Management Unit for Human Resources & Institutional Development, Research and Innovation (PMU-B) [Grant numbers B01F650006, BCG.659 and B05F650041] (NJ). SA is a recipient of Master Degree Student Research Assistantship from the Faculty of Medicine Ramathibodi hospital, and the Scholarships for Ph.D. Students from Mahidol University. The funders had no role in study design, data collection and analysis, decision to publish, or preparation of the manuscript.

**Competing interests:** The authors have declared that no competing interests exist.

**List of Abbreviations: CIN**: Cisplatin Induced Nephrotoxicity; **AKI**: Acute Kidney Injury; **AKD**: Acute Kidney Disease; **SNP**: Single Nucleotide Polymorphisms; *OCT2*: Organic Cation Transporter 2; *SLC22A2*: Solute Carrier Family 22 member 2; *ERCC1*: Excision Repair Cross-Complementing 1; *ERCC2*: Excision Repair Cross-Complementing 2; *EPHX1*: Epoxide Hydrolase 1; *XPD*: Xeroderma Pigmentosum group D; **DME**: Drug Metabolizing Enzyme; **PFS**: Progression-Free Survival; **OS**: Overall Survival; **CTCAE-AKI**: Common Criteria Terminology for Adverse Events - Acute Kidney Injury; **KDIGO**: Kidney Disease Improving Global Outcomes; **ICD-10**: 10th Revision of International Classification of Diseases; **T-Rex**: Thai Reference Exome Database; **MAF**: Minor Allele Frequency; **FDR**: False Discovery Rate; **BH-FDR**: Benjamini-Hochberg False Discovery Rate; **HWE**: Hardy-Weinberg Equilibrium.

## Introduction

Cisplatin (cis-diaminedichloroplatinum; CDDP; Platinol) is a chemotherapy drug, used for treatment of multiple cancers including head and neck squamous cell carcinoma (HNSCC), lung, and esophageal. Despite being an effective anticancer drug, its usage can be delayed or terminated because of associated toxicities such as neurotoxicity, gastrointestinal toxicity, ototoxicity and nephrotoxicity [1]. Platinum could be retained in the plasma for as long as 20 years after discontinuation of cisplatin-based chemotherapy [2,3], thus increasing concerns about the long-term nephrotoxicity risks [4].

Cisplatin-induced nephrotoxicity (CIN) occurs upon cisplatin accumulation in the renal proximal tubules. Its influx into the proximal tubule of nephron is mediated through organic cationic transporter 2 (*OCT2*/*SLC22A2*). Upon entering the renal tubules, complex interactions cause cisplatin to convert into a positive electrophile, which has higher affinity to DNA, forming DNA adducts, causing the DNA double helix to distort, thus hindering DNA synthesis and replication [5]. This DNA damage is detected and repaired by DNA repair pathways, i.e., nucleotide-excision repair (NER) and base excision repair (BER). Excision repair cross-complementing group 1 (*ERCC1*) and excision repair cross-complementing group 2 (*ERCC2*), also called X-ray repair complementing defective repair (*XPD*), are two of several proteins involved in these DNA repair mechanisms [6].

Clinically, CIN manifests in various forms, with Acute Kidney Injury (AKI) being the most common, occurring in 20–30% of patients [7]. Animal models of CIN demonstrates increased serum creatinine (SCr) and blood urea nitrogen, along with histopathological changes [8]. Though SCr is the main marker for renal function evaluation, it needs to be considered in conjunction with other factors such as age and sex. CIN is dose-dependent, with lower doses of cisplatin causing delayed kidney damage as compared to higher doses [9]. Cisplatin dosing regimens vary by cancer type, leading to differences in nephrotoxicity incidence. In lung and esophageal cancer treatments, ≥60 mg/m$^2$ every 3–4 weeks and 75 mg/m$^2$ on days 1 and 29 are administered, respectively [10,11]. Gradually, in HNSCC, weekly 40 mg/m$^2$ or 100mg/m$^2$ every 3 weeks is being administered concurrently with radiotherapy [12].

AKI may occur within 7 days of cisplatin administration, presented as increase in SCr, whereas Acute Kidney Disease (AKD) develops between 7 and 90 days of cisplatin initiation [13]. Prior studies have commonly used AKI as an endpoint for CIN based on different grading criteria; while the long-term or delayed effects of CIN are not widely studied. AKD, defined as renal damage occurring within 3 months of cisplatin administration, may prove to be a valuable endpoint for studying its delayed nephrotoxic effects [1, 14].

Cisplatin treatments have varying nephrotoxic outcomes. Studies have suggested that toxic and efficacious effects of cisplatin chemotherapy may result from genetic variation such as single nucleotide polymorphisms (SNP). Certainly, SNPs in drug metabolism, transport, and DNA repair genes have been shown to cause inter-individual differences in drug response and survival outcomes. Genetic variations in the drug metabolizing enzyme genes, *SLC22A2* and *EPHX1*, and the DNA repair genes, *ERCC1* and *ERCC2,* have been associated with CIN [4, 15].

Organic Cation Transporter 2 (*OCT2/SLC22A2*), part of the Solute Carrier Transport family, facilitates uptake of various substances into tubular cells. *SLC22A2* rs316019 has been linked to cisplatin nephrotoxicity [16,17]. Epoxide Hydrolase 1 (*EPHX1*) gene is a biotransformation enzyme involved in drug metabolism and excretion. It mainly catalyzes hydrolysis of reactive epoxides to transdihydrodiols, which may be further metabolized into polycyclic hydrocarbons diols, which are highly mutagenic. The dual roles of *EPHX1* in activation and detoxification of procarcinogens make it an important drug metabolism gene [18]. Variation rs1051740 causes the enzymatic activity to decrease approximately by 50%.

*ERCC1* and *ERCC2* or *XPD* are integral part of the NER pathway. SNPs in these genes - *ERCC1* rs11615 and rs3212986, and *ERCC2* rs13181 and rs1799793 - are linked to changes in the DNA repair process. Variations in these genes may cause a decrease in the helicase activity and DNA repair. *ERCC1* plays a role in incising the DNA damage site, and can cause a rate-limiting effect, while *ERCC2,* is involved in helicase unwinding the DNA at the damaged site. Hence, these SNPs may reduce the efficiency of DNA repair mechanisms, leading to greater platinum-DNA adduct formation and hindered nephron repair following cisplatin exposure [19].

Since SNP allele frequencies vary across populations, the association between these 6 SNPs and CIN in Thai cancer patients is unknown. This study aims to validate their associations with CIN in Thai patients with HNSCC, lung, and esophageal cancers.

## Materials and methods

### Patient selection

This study has been approved by the Human Research Ethics Committee, Faculty of Medicine Ramathibodi Hospital, Mahidol University, in compliance with the Declaration of Helsinki, protocol number COA. MURA2020/1109. Written informed consents were obtained from all participants.

Clinical and treatment data were obtained from an in-house observational head and neck cancer cohort database, Ramathibodi Tumor Registry, pharmacy database, and medical records included patient demographics, serum creatinine (SCr), estimated glomerular filtration rate (eGFR; CKD-EPI), cancer staging and comorbidities (ICD-10). All clinical information was verified by two independent medical oncologists. The clinical data were initially accessed on 15th July 2020. The authors had access to information that could identify individual participants during data collection; however, no individually identifiable data was reported in this study.

Solid tumor patients treated with cisplatin chemotherapy at the Faculty of Medicine, Ramathibodi Hospital, Mahidol University, Bangkok, Thailand were retrospectively identified. Criteria used for patient inclusion were: head & neck, lung and esophageal cancer patients, age ≥ 18 years, availability of blood samples in Ramathibodi Comprehensive Tumor Biobank collected between 1st January 2016–30th June 2020. Samples were collected from the biobank for research purposes starting from 17th March 2021. Patients with known chronic kidney disease as defined by The Kidney Disease Improving Global Outcomes (KDIGO) guideline before the start of cisplatin-based treatment were excluded [14].

For HNSCC cohort, in our prospective multidisciplinary observation study of HNSCC and NPC patients at Ramathibodi Hospital, Mahidol University, we have accrued eligible patients since 2016. The study enrolled patients with LA-HNSCC and NPC treated according to the standard of care at their treating physician's discretion. Enrollment was consistent, except during Covid-19 pandemic when the enrollment was hold, and subsequently resumed in 2021. Similarly, lung and esophageal cancer cohorts were also enrolled since 2016.

### Nephrotoxicity criteria

The study aimed to evaluate correlation of SNPs of interest and acute/subacute nephrotoxicity in Thai cancer patients who were treated with cisplatin. Acute and subacute nephrotoxicity were classified into acute kidney injury (AKI) and acute kidney disease (AKD) groups according to KDIGO criteria [14,20], which defines AKI as an increase in SCr by 50% within

7 days or 0.3 mg/dl (26.5 μmol/l) within 2 days or oliguria. AKD is defined as having AKI or GFR < 60 ml/min per 1.73 m² for < 3 months or a decrease in GFR by ≥35% or an increase in SCr by > 50% for < 3 months.

Laboratory values closest to and/or within 30 days prior to cisplatin start date were considered as the baseline to assess SCr and eGFR change [17]. Nephrotoxicity was evaluated as cases versus controls, i.e., AKI or Non-AKI and AKD or Non-AKD (Fig 1) shows the CONSORT diagram.

## Single nucleotide polymorphism (SNP) selection

Previous clinical studies and systematic reviews related to CIN-associated SNPs in different types of cancers and populations in PubMed from 2008–2020 were reviewed [4,15-17,21–29]. Only the candidate SNPs with minor allele frequency (MAF) ≥ 5% in Thai population were included. Finally, a total of 6 SNPs in 4 genes were selected. Allele frequencies in Thai and global populations were respectively obtained from the Thai Reference Exome Database (T-REx), which contains data from 1,092 Thai individuals (accessed on 25th February 2022), and from dbSNP (build 157) (accessed on 13th March 2025). *SLC22A2* rs316019 and *EPHX1* rs1051740 are involved in drug metabolism, while *ERCC1* rs11615 and rs3212986, and *ERCC2* rs13181 and rs1799793 play an important part in DNA repair mechanisms.

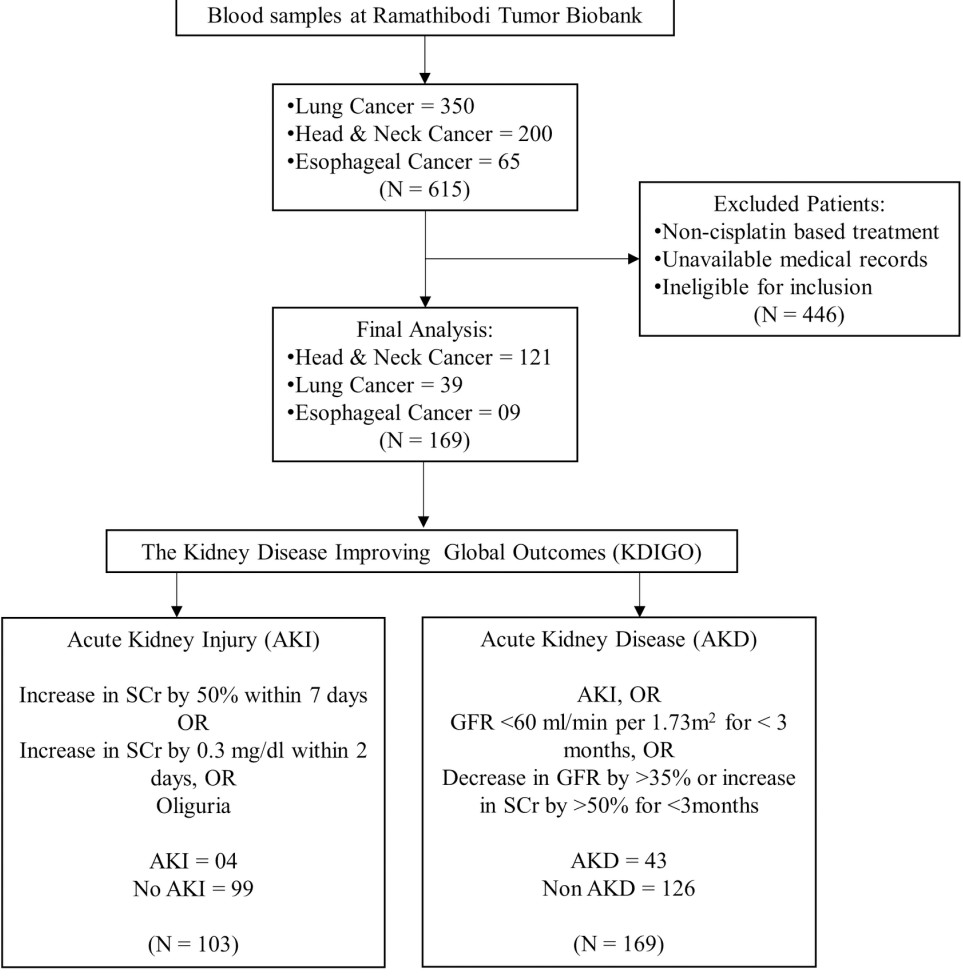

**Fig 1. CONSORT diagram.**

## DNA extraction and genotyping analysis

Genomic DNA (gDNA) was extracted from buffy coat samples using Gene All™ (Roche life sciences, Switzerland) for head and neck cancer cases, and GF-1 Blood DNA Extraction Kit (Vivantis Technologies, Malaysia) for lung and esophageal cancer cases, following each extraction kit's manufacturer's recommendation. gDNA was further quantified and qualified using NanoDrop™ spectrophotometer (ThermoFisher Scientific™, USA) followed by 1% agarose gel electrophoresis.

Genotyping was performed by TaqMan Drug Metabolizing Enzyme (DME) and Predesigned SNP genotyping assays using TaqMan™ GTXpress™ Master Mix (Applied Biosystems, USA), according to the manufacturer's protocol. TaqMan assay probe IDs of the 6 selected SNPs are as follows: *SLC22A2* (rs316019, C__3111809_20), *EPHX1* (rs1051740, C__14938_30), *ERCC1* (rs11615, C__2532959_20) and (rs3212986, C__2532948_10), *ERCC2* (rs13181, C__3145033_10) and (rs1799793, C__3145050_10). Genotyping and allelic discrimination plot analysis was performed on a ViiA™ 7 Real-Time PCR system (Applied Biosystems, USA).

Moreover, after obtaining the first set of TaqMan results, Sanger sequencing was done on randomly selected samples to confirm each genotype calls for all 6 SNPs S1 Table, which were then used as positive controls in the subsequent TaqMan runs. Primers designed and optimized by touchdown PCR protocol for Sanger sequencing are shown in the S2 Table. Sequencing chromatogram of all 6 SNPs were shown in S1 Fig.

## Statistical analyses

Patient demographics and clinical variables between cases and controls were compared using Pearson's chi-square or Fisher's exact test. Genotype frequencies were counted directly, while Hardy-Weinberg Equilibrium deviation in control group was assessed using *genhwi* command in STATA program. Power calculations showed that a sample size of 169 patients, assuming 25% nephrotoxicity, would provide our study with 80% power to identify a statistically significant association between the selected SNPs and AKD, with an alpha of 0.05 and an odds ratio of 2.8.

SNP association with cases and controls was analyzed using logistic regression. Models were adjusted by age and sex as covariates. Minor allele was considered as the risk allele and subsequently, 5 genetic inheritance models, namely co-dominant, dominant, recessive, over-dominant, and additive, were considered for analysis. For the additive and dominant genetic models, power calculations were conducted using STATA 14, power two proportions and RStudio 4.4.2 *genpwr* package, respectively. For the six SNPs analyzed, the estimated power varied with 53% being the highest.

Multivariate logistic regression was also performed to analyze potential factors affecting AKD. Multiple comparisons were corrected by Benjamini-Hochberg false discovery rate (FDR) corrected *P*-values, and additionally with Holm-Bonferroni correction.

Survival analyses focusing on overall survival (OS), progression free survival (PFS) and time-to-AKD were performed. OS was defined as the time of cancer diagnosis to death from any cause or last follow up of patients. OS was also performed in each cancer type. PFS was defined as the time of cisplatin initiation to the time of disease progression or death from any cause. PFS was performed only in head and neck cancer cohort because of limitation of data availability. Time to AKD was defined as the time of cisplatin initiation to the time of AKD appeared by KDIGO criteria. Cox regression analysis was performed to analyze the risk factors affecting mortality and time-to-AKD. The survival status was cross-checked with the National Security Death Index of Thailand on 23/3/2022.

The Kaplan-Meier curve and log-rank test were used to assess survival differences between AKD and non-AKD groups and the effect of polymorphisms on survival by dominant genetic model. Similarly, cumulative incidence of AKD by the 3 genotypes were presented using Kaplan-Meier curves. *P* < 0.05 was considered statistically significant. All statistical analysis and plots were performed using STATA 14.0 (College Station, TX: StataCorp LLC, USA).

## Results

### Patient characteristics

AKI was only developed in 4 (3.9%) out of 103 patients in our cohort, likely due to the improvement of hydration and supplementation regimens S3 Table. On the contrary, AKD occurred in 43 (25.4%) out of 169 patients. In this cohort, most patients were male (65.1%) and aged less than 65 years old (82.2%), whereas the median age being 57.4 years old (range 27.6–77.1). Hypertension was predominant among the comorbidities; 50 (29.6%), while diabetes, cerebrovascular and heart diseases were 15 (8.9%), 13 (7.7%) and 9 (5.3%), respectively. The majority of patients had head and neck cancer (72%). Squamous cell carcinoma (SCC) was the most common histologic subtype (73.4%). Fifty percent of all patients had stage IV disease followed by stage III (29%), stage II (14%), and stage I (7%) diseases. AKD patients had significantly lower mean eGFR ($P = <0.001$) and higher mean SCr ($P = 0.013$) as compared to non-AKD patients. Detailed demographics and clinicopathological data of the AKD and non-AKD groups are shown in Table 1. This cohort was then used for SNP association and survival analyses.

### Associations between the candidate SNPs and cisplatin-induced AKD across all cancer types

Genetic polymorphism association with AKD was performed in the 169 cancer patients. All candidate SNPs have minor allele frequencies ≥ 0.05 in Thai population and there was no significant deviation from Hardy-Weinberg Equilibrium

**Table 1. Clinical characteristics of AKD and Non-AKD patients after receiving cisplatin chemotherapy.**

| Factors | Overall (n = 169) | AKD (n = 43) (25.4%) | Non-AKD (n = 126) (74.6%) | P value |
|---|---|---|---|---|
| **Sex** | | | | |
| Male | 110 (65.1%) | 25 (58.1) | 85 (67.5) | 0.268[a] |
| Female | 59 (34.9%) | 18 (41.9) | 41 (32.5) | |
| **Median Age (range)** | | | | |
| Age | 57.4 (27.6-77.1) | 59.8 (38.7-73.9) | 56.2 (27.6-77.1) | 0.528[a] |
| <65 | 139 (82.2%) | 34 (79.1) | 105 (83.3) | |
| ≥65 | 30 (17.8%) | 9 (20.9) | 21 (16.7) | |
| **Comorbidity (ICD-10)** | | | | |
| Hypertension | 50 (29.6%) | 17 (39.5) | 33 (26.2) | 0.098[a] |
| Diabetes Mellitus | 15 (8.9%) | 4 (9.3) | 11 (8.7) | 1.000[b] |
| Cerebrovascular Disease | 13 (7.7%) | 4 (9.3) | 9 (7.1) | 0.741[b] |
| Heart Disease | 9 (5.3%) | 2 (4.7) | 7 (5.6) | 1.000[b] |
| **Cancer Type** | | | | |
| HNSCC | 121 (71.6%) | 27 (62.8) | 94 (74.6) | 0.333[a] |
| Lung | 39 (23.1%) | 13 (30.2) | 26 (20.6) | |
| Esophagus | 9 (5.3%) | 3 (6.9) | 6 (4.8) | |
| **Histology** | | | | |
| SCC | 124 (73.4%) | 29 (67.4) | 95 (75.4) | 0.308[a] |
| Non-SCC | 45 (26.6%) | 14 (32.6) | 31 (24.6) | |
| **Staging** | | | | |
| I | 12 (7.1%) | 4 (9.3) | 8 (6.4) | 0.668[a] |
| II | 23 (13.6%) | 5 (11.6) | 18 (14.3) | |
| III | 49 (29.0%) | 10 (23.3) | 39 (30.9) | |
| IV | 85 (50.3%) | 24 (55.8) | 61 (48.4) | |
| **Baseline mean eGFR (+/-SD) ml/min/1.73m²** | 98.77 ± 14.78 | 90.90 ± 15.95 | 101.45 ± 13.40 | **<0.001*** |
| **Baseline mean SCr (+/-SD) mg/dL** | 0.75 ± 0.17 | 0.81 ± 0.20 | 0.73 ± 0.15 | **0.013*** |

[a]Calculated using Pearson's chi-square test. [b] calculating using Fisher's exact test. * Statistically significant *P* value < 0.05. SCC, Squamous Cell Carcinoma. HNSCC, Head and Neck Squamous Cell Carcinoma.

(HWE) Table 2. Genetic association between each SNP and AKD was studied using logistic regression in 5 genetic models: co-dominant, dominant, recessive, over-dominant and additive. Association analysis was done for both unadjusted and adjusted models. Unadjusted model showed TC genotype of *EPHX1* rs1051740 was significantly associated with increased risk of AKD in co-dominant; TC vs CC [OR 2.894, 95% CI 1.091–7.680, *P* = 0.033], and over-dominant genetic models; TC vs CC + TT [OR 2.793, 95% CI 1.333–5.851, *P* = 0.006].

Similarly, after adjusting the model with age and sex, TC genotype of *EPHX1* rs1051740 remained significantly associated with increased AKD risk in co-dominant and over-dominant models [OR 3.354, 95% CI 1.230–9.148, *P* = 0.018] and [OR 3.137, 95% CI 1.450–6.785, *P* = 0.004], respectively. However, both associations did not remain significant after correcting for multiple comparisons using FDR. None of the other 5 SNPs were associated with AKD in any of the genetic model. Table 3 summarizes the association analysis and genotype frequencies in AKD and Non-AKD groups.

Upon adjusting the model with other clinical factors such as hypertension, diabetes, heart diseases and cerebrovascular diseases and histological types, also showed TC genotype of *EPHX1* rs1051740 to be significantly associated with AKD in both co-dominant and over-dominant models, although the associations were no longer significant after FDR and Holm-Bonferroni corrections.

Additionally, a multivariate logistic regression analysis was performed using *EPHX1* rs1051740 genotype, the statistically significant variables, and potential AKD-associated clinical factors to analyze their effect on AKD. We showed that the eGFR baseline cutoff <90 ml/min/1.73m2 and TC genotype of *EPHX1* rs1051740 were significantly associated with an increased risk of cisplatin-induced AKD among cancer patients. However, no significant association was found between age, hypertension and diabetes mellitus, as summarized in Table 4.

## Cumulative incidence of AKD over time in association with the candidate SNPs

Fig 2A showed the cumulative incidence of AKD over a 90-day period across our patient cohort. Cumulative incidence of AKD over time in association with each SNP (co-dominant model) was subsequently analyzed by Kaplan-Meier method (Figs 2B-2G). The only SNP that showed statistically significant association with the increased cumulative incidence of AKD over time was *EPHX1* rs1051740 (*P* = 0.021) when comparing TC with CC genotype (Fig 2C). *SLC22A2* rs316019, *ERCC1* rs11615 and rs3212986, as well as *ERCC2* rs13181 and rs1799793 did not exhibit significant findings when

**Table 2. Information and allele frequencies of the 6 selected SNPs in Thai population.** Allele frequencies were obtained from dbSNP (build 157; https://www.ncbi.nlm.nih.gov/snp/), and from the T-REx database [30].

| Gene | SNP | SNP Information | Global Frequencies | | T-REx Frequencies | | Data Allele Frequencies | | HWE *P* value |
|---|---|---|---|---|---|---|---|---|---|
| *SLC22A2* | **rs316019** | Missense c.808G > T (p.270Ala > Ser) | C | A | C | A | C | A | 0.187 |
| | | | 0.895 | 0.104 | 0.890 | 0.110 | 0.828 | 0.171 | |
| *EPHX1* | **rs1051740** | Missense c.337T > C (p.Tyr113His) | C | T | C | T | C | T | 0.331 |
| | | | 0.295 | 0.704 | 0.504 | 0.496 | 0.503 | 0.496 | |
| *ERCC1* | **rs11615** | Synonymous c.354T > C (p.Asn118Asn) | G | A | G | A | G | A | 0.482 |
| | | | 0.414 | 0.585 | 0.697 | 0.303 | 0.600 | 0.400 | |
| *ERCC1* | **rs3212986** | Synonymous 3' UTR c*197G > T | C | A | C | A | C | A | 0.261 |
| | | | 0.745 | 0.254 | 0.664 | 0.336 | 0.720 | 0.279 | |
| *ERCC2* | **rs13181** | Missense c.2251A > C (p.751lys > Gln) | T | G | T | G | T | G | 0.910 |
| | | | 0.642 | 0.000 | 0.901 | 0.099 | 0.910 | 0.089 | |
| *ERCC2* | **rs1799793** | Missense c.934G > A (p.Asp312Asn) | C | T | C | T | C | T | 0.889 |
| | | | 0.684 | 0.315 | 0.937 | 0.063 | 0.942 | 0.057 | |

HWE *P*, Hardy-Weinberg Equilibrium *P* value of the non-AKD (control) group.

**Table 3. Association of the selected genetic polymorphisms with cisplatin-induced AKD. Association analyses were performed by logistic regression using 5 genetic models. Unadjusted OR was calculated between AKD outcome and SNP. Model was then adjusted with age and sex variables.**

| SNP | Model | Genotype | AKD (%) | Non-AKD (%) | Unadjusted OR (95% CI) | P value | BH-FDR P value | Holm-Bonferroni P value | Adjusted OR (95% CI) | P value Adj | BH-FDR P value Adj | Holm-Bonferroni P value Adj |
|---|---|---|---|---|---|---|---|---|---|---|---|---|
| **SLC22A2 rs316019** | Co-dominant | CC | 32 (74.4) | 101 (80.1) | 1 | | | | 1 | | | |
| | | AC | 11 (25.6) | 22 (17.5) | 1.578 (0.691 - 3.603) | 0.279 | 0.721 | 0.003 | 1.532 (0.656 - 3.578) | 0.323 | 0.738 | 0.003 |
| | | AA | 0 (0.0) | 3 (2.4) | - | - | - | - | - | - | - | - |
| | Dominant | AA+AC | 11 (25.6) | 25 (19.8) | 1.388 (0.615 - 3.131) | 0.429 | 0.881 | 0.003 | 1.362 (0.591 - 3.139) | 0.468 | 0.881 | 0.003 |
| | Recessive | CC+AC | 43 (100) | 123 (97.6) | 1 | | | | 1 | | | |
| | | AA | 0 (0.0) | 3 (2.4) | - | - | - | - | - | - | - | - |
| | Over-dominant | CC+AA | 32 (74.4) | 104 (82.5) | 1 | | | | 1 | | | |
| | | AC | 11 (25.6) | 22 (17.5) | 1.625 (0.712 - 3.708) | 0.249 | 0.693 | 0.002 | 1.576 (0.675 - 3.679) | 0.293 | 0.721 | 0.003 |
| | Log-Additive | | | | 1.165 (0.562 - 2.415) | 0.681 | 0.935 | 0.005 | 1.157 (0.546 - 2.452) | 0.702 | 0.935 | 0.006 |
| **EPHX1 rs1051740** | Co-dominant | CC | 6 (14.0) | 33 (26.2) | 1 | | | | 1 | | | |
| | | TC | 30 (69.8) | 57 (45.2) | 2.894 (1.091 - 7.680) | 0.033* | 0.526 | 0.002 | 3.354 (1.230 - 9.148) | 0.018* | 0.384 | 0.002 |
| | | TT | 7 (16.2) | 36 (28.6) | 1.069 (0.325 - 3.509) | 0.912 | 0.941 | 0.017 | 1.138 (0.336 - 3.843) | 0.835 | 0.941 | 0.010 |
| | Dominant | TT+TC | 37 (86.0) | 93 (73.8) | 2.188 (0.846 - 5.655) | 0.106 | 0.526 | 0.002 | 2.459 (0.931 - 6.493) | 0.069 | 0.526 | 0.002 |
| | Recessive | CC+TC | 36 (83.7) | 90 (71.4) | 1 | | | | 1 | | | |
| | | TT | 7 (16.3) | 36 (28.6) | 0.486 (0.198 - 1.192) | 0.115 | 0.526 | 0.002 | 0.475 (0.188 - 1.199) | 0.115 | 0.526 | 0.002 |
| | Over-dominant | CC+TT | 13 (30.2) | 69 (54.8) | 1 | | | | 1 | | | |
| | | TC | 30 (69.8) | 57 (45.2) | 2.793 (1.333 - 5.851) | 0.006* | 0.192 | 0.002 | 3.137 (1.450 - 6.785) | 0.004* | 0.192 | 0.002 |
| | Log-Additive | | | | 0.998 (0.607 - 1.642) | 0.996 | 0.996 | 0.050 | 1.039 (0.624 - 1.730) | 0.881 | 0.941 | 0.017 |
| **ERCC1 rs11615** | Co-dominant | GG | 21 (48.8) | 59 (46.8) | 1 | | | | 1 | | | |
| | | AG | 17 (39.5) | 57 (45.2) | 0.837 (0.401 - 1.748) | 0.638 | 0.935 | 0.004 | 0.853 (0.401 - 1.815) | 0.680 | 0.935 | 0.005 |
| | | AA | 5 (11.6) | 10 (8.0) | 1.404 (0.430 - 4.587) | 0.574 | 0.935 | 0.004 | 1.081 (0.315 - 3.706) | 0.900 | 0.941 | 0.025 |
| | Dominant | AA+AG | 22 (51.2) | 67 (53.2) | 0.922 (0.461 - 1.844) | 0.820 | 0.941 | 0.008 | 0.892 (0.437 - 1.821) | 0.755 | 0.935 | 0.007 |
| | Recessive | GG+AG | 38 (88.4) | 116 (92.1) | 1 | | | | 1 | | | |
| | | AA | 5 (11.6) | 10 (7.9) | 1.526 (0.490 - 4.745) | 0.465 | 0.881 | 0.003 | 1.164 (0.357 - 3.796) | 0.800 | 0.941 | 0.008 |
| | Over-dominant | GG+AA | 26 (60.5) | 69 (54.8) | 1 | | | | 1 | | | |
| | | AG | 17 (39.5) | 57 (45.2) | 0.791 (0.391 - 1.601) | 0.516 | 0.917 | 0.004 | 0.841 (0.407 - 1.737) | 0.642 | 0.935 | 0.004 |
| | Log-Additive | | | | 1.041 (0.609 - 1.779) | 0.883 | 0.941 | 0.013 | 0.966 (0.559 - 1.667) | 0.901 | 0.941 | 0.050 |
| **ERCC1 rs3212986** | Co-dominant | CC | 20 (46.5) | 62 (49.2) | 1 | | | | 1 | | | |

(Continued)

| SNP | Model | Genotype | AKD (%) | Non-AKD (%) | Unadjusted OR (95% CI) | P value | BH-FDR P value | Holm-Bonferroni P value | Adjusted OR (95% CI) | P value Adj | BH-FDR P value Adj | Holm-Bonferroni P value Adj |
|---|---|---|---|---|---|---|---|---|---|---|---|---|
| | | CA | 17 (39.5) | 49 (38.8) | 1.075 (0.509 - 2.270) | 0.849 | 0.941 | 0.010 | 1.136 (0.526 - 2.451) | 0.744 | 0.935 | 0.006 |
| | | AA | 6 (14.0) | 15 (12.0) | 1.240 (0.424 - 3.624) | 0.694 | 0.935 | 0.006 | 1.329 (0.442 - 3.988) | 0.612 | 0.935 | 0.004 |
| | Dominant | AA+AC | 23 (53.5) | 64 (50.8) | 1.114 (0.556 - 2.229) | 0.760 | 0.935 | 0.007 | 1.182 (0.580 - 2.408) | 0.645 | 0.935 | 0.004 |
| | | CC+CA | 37 (86.0) | 111 (88.1) | 1 | | | | 1 | | | |
| | Recessive | AA | 6 (14.0) | 15 (11.9) | 1.200 (0.433 - 3.318) | 0.725 | 0.935 | 0.006 | 1.256 (0.442 - 3.569) | 0.668 | 0.935 | 0.005 |
| | | CC+AA | 26 (60.5) | 77 (61.1) | 1 | | | | 1 | | | |
| | Over-dominant | CA | 17 (39.5) | 49 (38.9) | 1.027 (0.505 - 2.086) | 0.940 | 0.955 | 0.025 | 1.071 (0.516 - 2.222) | 0.853 | 0.941 | 0.013 |
| | Log-Additive | | | | 1.103 (0.671 - 1.810) | 0.668 | 0.935 | 0.005 | 1.148 (0.692 - 1.905) | 0.592 | 0.935 | 0.003 |
| *ERCC2* rs13181 | Co-dominant | TT | 37 (86.0) | 95 (75.4) | 1 | - | - | | 1 | - | - | |
| | | TG | 6 (14.0) | 29 (23.0) | 0.531 (0.203 - 1.384) | 0.195 | 0.594 | 0.002 | 0.470 (0.171 - 1.291) | 0.143 | 0.551 | 0.002 |
| | | GG | 0 (0.0) | 2 (1.6) | - | - | - | | - | - | - | |
| | Dominant | GG+TG | 6 (14.0) | 31 (24.6) | 0.496 (0.191 - 1.288) | 0.150 | 0.551 | 0.002 | 0.443 (0.162 - 1.210) | 0.112 | 0.526 | 0.002 |
| | | TT+TG | 43 (100) | 124 (98.4) | 1 | | | | 1 | | | |
| | Recessive | GG | 0 (0.0) | 2 (1.6) | - | - | - | | - | - | - | |
| | | TT+GG | 37 (86.1) | 97 (77.0) | 1 | | | | 1 | | | |
| | Over-dominant | TG | 6 (13.9) | 29 (23.0) | 0.542 (0.208 - 1.412) | 0.210 | 0.611 | 0.002 | 0.481 (0.175 - 1.319) | 0.155 | 0.551 | 0.002 |
| | Log-Additive | | 29 (23.0) | | 0.491 (0.196 - 1.230) | 0.129 | 0.550 | 0.002 | 0.441 (0.166 - 1.167) | 0.099 | 0.526 | 0.002 |
| *ERCC2* rs1799793 | Co-dominant | CC | 39 (90.7) | 103 (81.7) | 1 | | | | 1 | | | |
| | | CT | 3 (7.0) | 22 (17.5) | 0.360 (0.103 - 1.271) | 0.113 | 0.526 | 0.002 | 0.340 (0.091 - 1.261) | 0.107 | 0.526 | 0.002 |
| | | TT | 1 (2.3) | 1 (0.8) | 2.641 (0.161 - 43.265) | 0.496 | 0.907 | 0.003 | 4.056 (0.226 - 72.726) | 0.342 | 0.742 | 0.003 |
| | Dominant | TT+CT | 4 (9.3) | 23 (18.3) | 0.459 (0.149 - 1.413) | 0.175 | 0.586 | 0.002 | 0.451 (0.140 - 1.453) | 0.183 | 0.586 | 0.002 |
| | | CC+CT | 42 (97.7) | 125 (99.2) | 1 | | | | 1 | | | |
| | Recessive | TT | 1 (2.3) | 1 (0.8) | 2.976 (0.182 - 48.638) | 0.444 | 0.881 | 0.003 | 4.786 (0.265 - 86.183) | 0.288 | 0.721 | 0.002 |
| | | CC+TT | 40 (93.0) | 104 (82.5) | 1 | | | | 1 | | | |
| | Over-dominant | CT | 3 (7.0) | 22 (17.5) | 0.354 (0.100 - 1.250) | 0.107 | 0.526 | 0.002 | 0.331 (0.089 - 1.224) | 0.288 | 0.526 | 0.002 |
| | Log-Additive | | | | 0.600 (0.225 - 1.599) | 0.308 | 0.730 | 0.003 | 0.609 (0.217 - 1.712) | 0.348 | 0.742 | 0.003 |

OR, Odds Ratio. 95% CI, 95% Confidence Interval. BH-FDR, Benjamini-Hochberg False Discovery Rate. Adj, adjusted P value. * Statistically significant P value <0.05.

**Table 4. Multivariate logistic analysis to analyze potential factors affecting AKD.**

| Factors | | Multivariate Analysis | | |
|---|---|---|---|---|
| | | Odds Ratio | 95% Confidence Interval | *P* value |
| Age ≥ 65 | | 1.40 | 0.50 - 3.92 | 0.513 |
| Hypertension | | 1.51 | 0.65 - 3.51 | 0.331 |
| Diabetes Mellitus | | 1.00 | 0.25 - 4.01 | 0.993 |
| eGFR Baseline (ml/min/1.73m²) <90 | | 3.30 | 1.41 - 7.71 | **0.006*** |
| *EPHX1* rs1051740 | TC | 2.79 | 1.01 - 7.86 | **0.046*** |
| | TT | 1.18 | 0.34 - 4.06 | 0.791 |

* Statistically significant *P* value < 0.05.

compared against major homozygote genotypes. This result was in line with our prior association analyses between the candidate SNPs and AKD. In addition, multivariate Cox regression analysis using significant variables were performed and identified age, baseline eGFR, SCr and individuals carrying TC genotypes of the *EPHX1* rs1051740 SNP as significant in univariate analysis [HR 1.040, CI 95% 1.006–1.076, *P* = 0.019; HR 0.954, CI 95% 0.934–0.974, *P* = <0.001; HR 14.490, CI 95% 2.167–96.873, *P* = 0.006; HR 2.433, 1.012–5.847, *P* = 0.047, respectively]. However, none of the variables analyzed were identified as independent factors of time-to-AKD after multivariate Cox analysis. The associated risk factors of time-to-AKD are listed in S7 Table.

### Associations between the candidate SNPs and cisplatin-induced AKD in each cancer type

Additionally, SNP association with AKD was analyzed in the head and neck cancer-only and the lung cancer-only cohorts, consisting of 121 and 39 patients, respectively. In head and neck cancer-only cohort, TC genotype of *EPHX1* rs1051740 was significantly associated with an increased risk of AKD in co-dominant [OR 13.333, 95% CI 1.684–105.533, *P* = 0.014] and over-dominant models [OR 4.029, 95% CI 1.552–10.455, *P* = 0.004]. Dominant model TT + TC vs CC also showed a significantly increased risk of AKD [OR 9.941, 95% CI 1.282–77.051, *P* = 0.028] S4 Table. Of note, bootstrap estimation was performed for the head and neck cancer-only cohort to correct for the wide confidence interval, which could lead to overestimation of the results S4 Table.

Moreover, the same finding was obtained when combining the 9 patients with esophageal cancer with the head and neck cancer-only cohort, or adjusting the logistic model with age and sex S5 Table. In the lung cancer-only cohort, none of the 6 SNPs showed any association with AKD, similar result was observed upon adjusting the logistic models with age & sex S6 Table.

### Survival analyses of the cancer patients in association with cisplatin-induced AKD or the candidate SNPs

In order to clarify whether the AKD status or the candidate CIN-associated SNPs may affect the survival outcome of the Thai cancer patients in our cohort, first, we analyzed the association between overall survival (OS) and patients with and without AKD using Kaplan-Meier curve. There was no significant difference in the OS between patients with and without AKD after a median follow-up time of 3.24 years [log-rank *P* = 0.094] (Fig 3A). Secondly, OS association with the 6 selected SNPs was studied under dominant model. SNPs in *SLC22A2*, *EPHX1* and *ERCC1* showed no associations with OS. Only *ERCC2* rs13181 and rs1799793 were significantly associated with poor OS, i.e., genotypes GG + TG of *ERCC2* rs13181 (*P* = 0.002) and TT + CT of *ERCC2* rs1799793 (*P* = 0.004), as compared to the homozygote genotypes (Figs 3B-3G).

Furthermore, no difference in OS was found between AKD and non-AKD groups when using only head and neck or lung cancer cohorts S2A and S3A Figs. For SNP association analysis, in the head and neck cancer only cohort, no

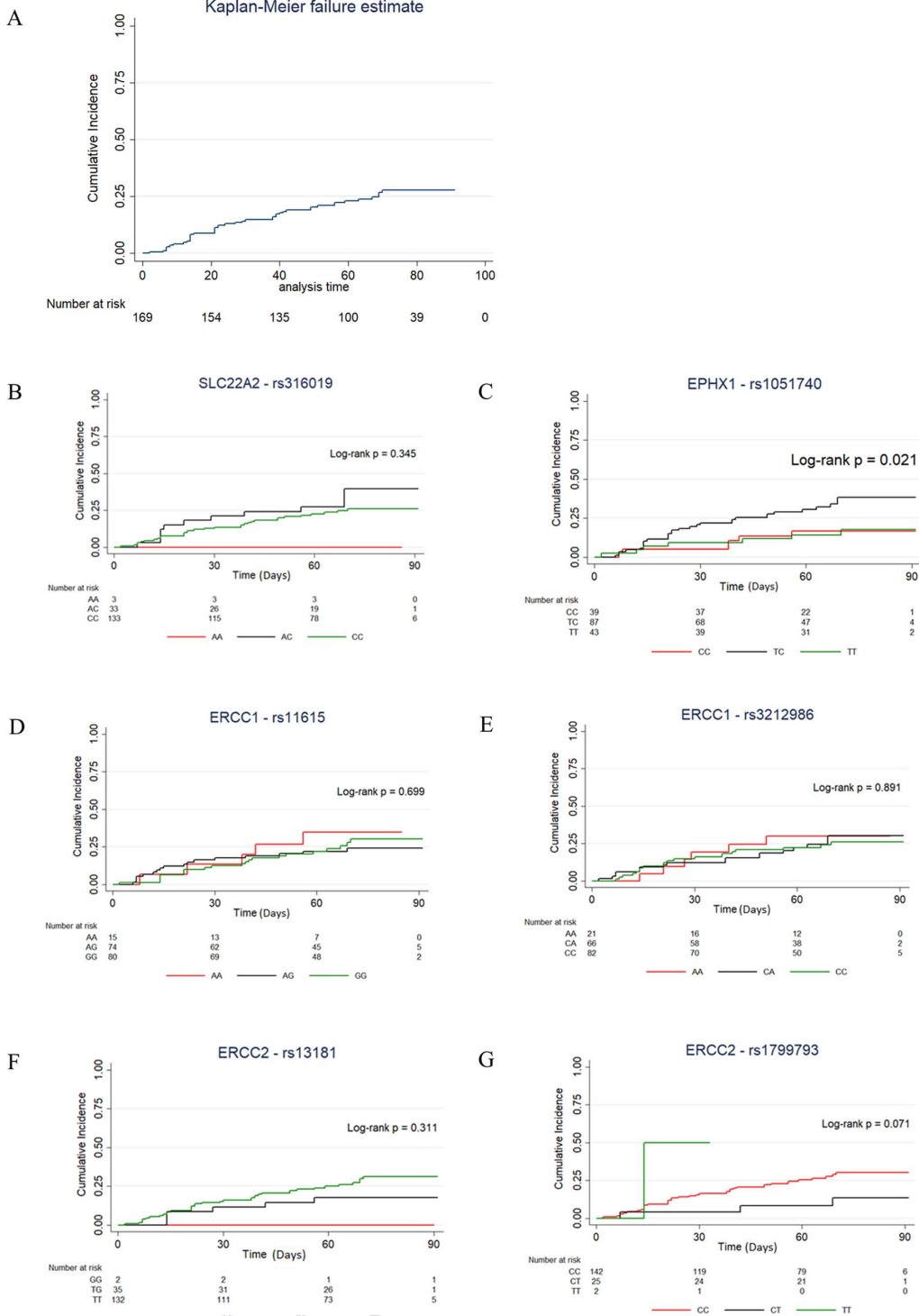

**Fig 2. Cumulative incidence of AKD in association with the 6 candidate SNPs.** (A) Cumulative incidence of AKD event in our patient cohort. Kaplan Meier curves are shown where the y-axis denotes cumulative incidence of patients experiencing AKD while the x-axis denotes days after the start of cisplatin. Ninety-day period of AKD is shown. Cumulative incidence of AKD in association with (B) *SLC22A2* rs316019, (C) *EPHX1* rs1051740, (D) *ERCC1* rs11615, (E) *ERCC1* rs3212986, (F) *ERCC2* rs13181 and (G) *ERCC2* rs1799793. $P < 0.05$ was considered significant.

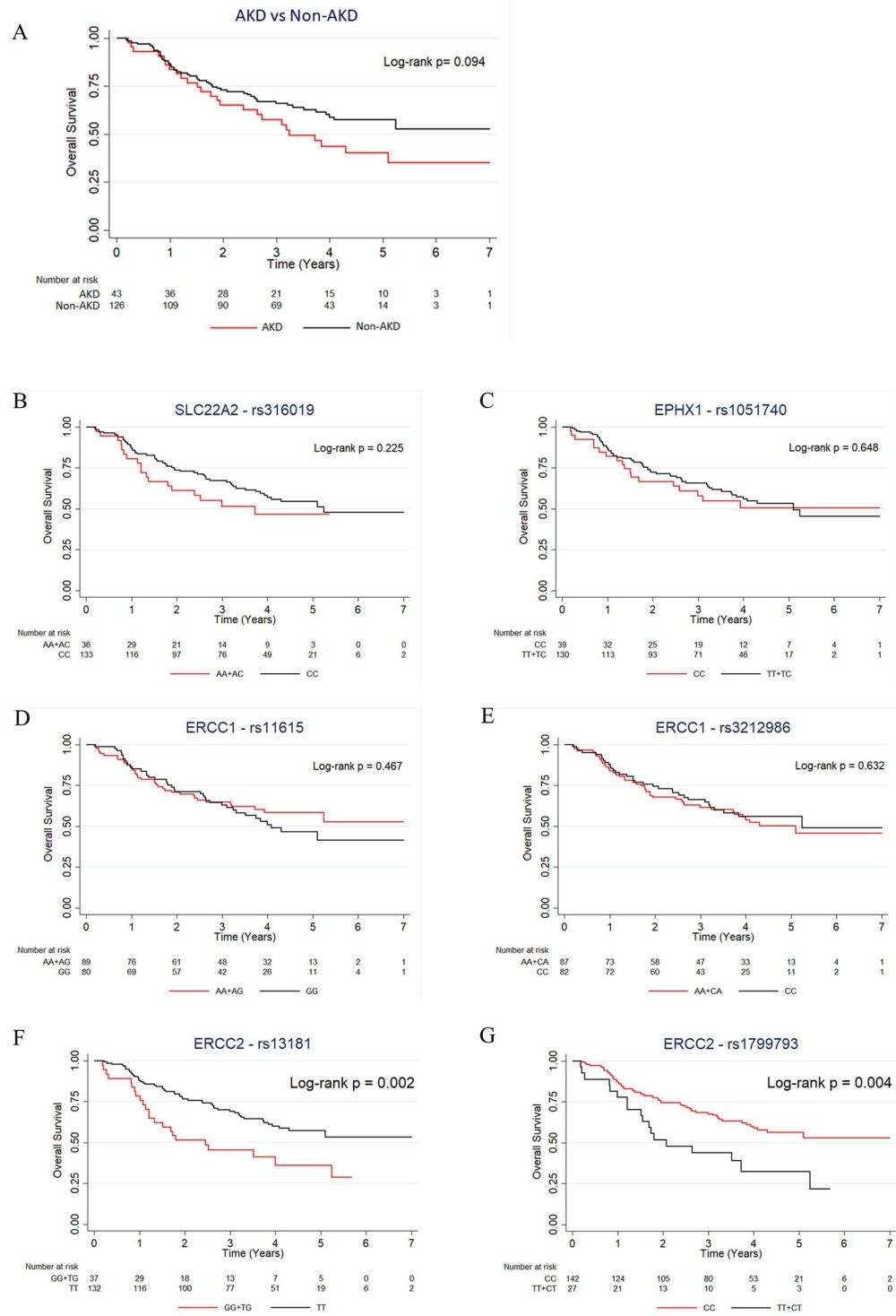

**Fig 3. Overall survival analyses of the 169 cancer patients.** (A) Overall survival in patients with and without AKD at a median follow-up time of 3.24 years by dominant genetic model. Overall survival in relation to SNP genotypes (B) *SLC22A2* rs316019, (C) *EPHX1* rs1051740, (D) *ERCC1* rs11615, (E) *ERCC1* rs3212986, (F) *ERCC2* rs13181 and (G) *ERCC2* rs1799793. *P* < 0.05 was considered significant.

significant association was observed in *SLC22A2, EPHX1* and *ERCC1* SNPs S2B – S2E Figs. Only genotypes GG + TG of *ERCC2* rs13181 and TT + TC of *ERCC2* rs1799793 were significantly associated with inferior survival outcome, similar to the finding when using all cases S2F and S2G Figs respectively. In contrast, the lung cancer only cohort did not show significant difference in OS in relation to any SNPs S3B –S3G Figs. Multivariate Cox regression was also performed to analyze the significant variables and potential risk factors for OS, and identified male, diabetes mellitus, heart disease, G allele on *ERCC2* rs13181 and T allele on rs1799793 tended to have an increased risk of mortality [HR 1.815, 95% CI 1.077–3.058, *P* = 0.025; HR 2.219, 95% CI 1.133–4.343, *P* = 0.020; HR 2.674, CI 95% 1.222–5.851, *P* = 0.014; HR 2.080, CI 95% 1.270–3.407, *P* = 0.004; HR 2.107, CI 95% 1.238–3.586, *P* = 0.006, respectively] whereas NSCC had decreased mortality risk [HR 0.508, CI 95% 0.279–0.925, *P* = 0.027] in the univariate Cox regression model, but this did not retain statistical significance in the multivariate Cox proportional hazards model. List of associated factors affecting mortality are given in S8 Table.

Lastly, PFS in association with AKD and SNPs was only analyzed in the 121 patients with head and neck cancer whose progression data was available. Our results demonstrated no association between PFS and kidney function S4A Fig and no association between PFS and *SLC22A2, EPHX1* and *ERCC1* SNPs S4B – S4E Figs. However, there were significant associations between PFS and *ERCC2* SNPs, rs13181 (*P* = 0.017) and rs1799793 (*P* = 0.023) S4F– S4G Figs.

## Discussion

In this study, we analyzed 6 SNPs in drug metabolizing enzyme and DNA repair genes and assessed their associations with CIN in Thai cancer patients. The effect of these SNPs on time-to-AKD, PFS, and OS were also investigated. A recent study by Patimarattananan et al. in Thai locally advanced Head and Neck Squamous Cell Carcinoma (HNSCC) patients who received chemoradiotherapy with cisplatin reported an incidence of AKI as 13.4% and AKD as 27.9% [1]. The incidence of AKI in our retrospective cohort, which was enrolled between 2016–2020, was only 3.9%, while AKD was 25.4%. The use of higher cisplatin doses in the past regimens may be one of the reasons for a higher AKI incidence seen in Patimarattananan et al. The recent improvement of hydration and supplementation regimens may also contribute to lower AKI incidence. Additionally, our cohort consisted of multiple types of cancer patients with various staging, influencing treatment methods, i.e., systemic chemotherapy or concurrent chemoradiation with cisplatin, and the laboratory testing timepoints, which could result in the underestimation of AKI incidence by KDIGO guideline. Therefore, we focused on AKD as an endpoint of CIN in our study.

AKD is a relatively new clinical classification representing patients who do not meet the definition of AKI or chronic kidney disease (CKD) and in whom pathophysiologic processes are still ongoing after AKI [31–33]. Evaluation of AKD in terms of CIN is vital for a comprehensive patient care. Patients developing AKI may progress to AKD, which poses persistent risks such as mortality and worsened kidney function [34]. Among patients with AKD post-hospital discharge, the long-term risk includes development of CKD [32]. Patients experiencing AKD along with co-morbidities may be exposed to risk factors that may enhance AKD severity. A study has shown that in-ICU and in-hospital mortality rates were significantly higher in intensive care unit patients who developed AKD than those with AKI. Similar to our patient's cohort, male sex, and lower baseline eGFR were among AKD progression factors [35]. Furthermore, a recent meta-analysis highlighted higher mortality risk in the AKD group compared to the non-AKD group [34]. These reports also highlight the heterogeneity among studies, necessitating further research to explore AKD outcomes in different populations. Hence, these findings underscore the need for studying AKD in addition to AKI, allowing for its close monitoring, early identification, and appropriate therapeutic interventions.

We found *EPHX1* rs1051740 significantly associated with cisplatin-induced AKD in co-dominant and over-dominant model, particularly in TC pattern. To our knowledge, no prior studies have performed AKD association analysis with genetic polymorphisms. Multivariate analysis, identified lower baseline eGFR less than 90 ml/min/1.73m$^2$ as another factor influencing AKD development after cisplatin treatment. However, comorbidities including hypertension, diabetes mellitus

and age did not show significant associations in our study, whereas Patimarattanan et al. suggests that hypertension was significantly associated with AKD development. In line with the same study [1], kidney function after cisplatin treatment did not impact the OS in our study.

No consensus has been reached regarding SNPs associated with CIN, partly due to significant variations in minor allele frequencies across populations. Previous studies have shown association of these 6 selected SNPS with CIN using AKI as the endpoint. Notably, the minor allele frequency of *EPHX1* rs1051740 differs in the Thai population compared to other populations [36]. Most commonly the reference allele is T, whose expression as homozygous TT genotype is associated with faster metabolism than CC. However, in Thai population, T is the minor allele, with a frequency of 0.49. The *EPHX1* rs1051740 (c.337T > C), is one of the 2 widely studied polymorphisms of the *EPHX1* gene. This variant results in a tyrosine-to- histidine substitution at position 113, leading to a 50% reduction in enzyme activity [37]. Decreased enzymatic activity may reduce the detoxification, increasing the formation of highly reactive metabolites, making the cells more susceptible to challenges.

*EPHX1* has previously been studied in various cancers. Association of rs1051740 and esophageal cancer remains inconsistent [38–40]. Previous meta-analysis studies show a significant increase in lung cancer risk among Asians with CC genotype or C allele in their subgroup analysis by ethnicity [41–43]. Lee et al. demonstrated a link between genetically lower epoxide hydrolase activity and an increased risk of tobacco related cancers among smokers [44]. Another meta-analysis found an association between rs1051740 polymorphism and HNSCC risk in population-based studies. [18]. While *EPHX1* is well-studied regarding cancer risk, few studies have explored its relationship with CIN. To date, only Khrunin et al. have reported a higher frequency of CIN in ovarian cancer patients with the rs1051740 heterozygous genotype in a Russian population [15].

Our association analysis results show *EPHX1* rs1051740 heterozygous genotype to be associated with AKD in our combined three-cancer-type cohort. This SNP was also significantly associated with AKD in the head and neck cancer-only cohort, although the confidence interval (CI) was wide, likely due to the smaller sample size. This association may hold significance, as AKD has not previously been studied with any SNP. This association may also be relevant because the heterozygous enzyme *EPHX1* may act as an intermediate metabolizer causing delayed elimination of cisplatin from the body. Since AKD also causes a delayed effect on kidneys, it may be a scenario where delayed metabolism of cisplatin may cause a prolonged nephrotoxic outcome. AKD time period is 3 months from the start date of cisplatin, nonetheless it is possible that there may be other factors unrelated to cisplatin that affects the kidneys. Patients undergoing multiple radiological testing might have a negative affect by the dyes used. Another possibility of having heterozygous genotype association may be because of sampling error, i.e., our cohort might lack sufficient patients with CC genotype.

*ERCC1* and *ERCC2* are key components of the NER pathway, activated by the formation of DNA adducts. Genetic variations in these genes may influence DNA repair capacity. Synonymous variants in the 3' UTR region such as *ERCC1* rs11615 and rs3212986, can affect protein stability, folding, structure, and expression [19, 45]. Previous studies by Khrunin et al. and Tzvetkov et al. reported an increased risk of CIN in ovarian and other cancers related to *ERCC1* rs11615 and rs3212986 in Caucasian [25,26]. However, Zazuli et al. found *ERCC1* rs3212986 to reduce risk of CIN in Caucasian testicular cancer patients [17]. In East-Asians, Chen et al. found no significant association between CIN and *ERCC1* rs11615 in Chinese lung cancer patients [46]. The association of *ERCC2* with CIN has also been widely studied, but results vary due to differing endpoints and populations. Studies have shown an increased risk of CIN in Caucasian osteosarcoma and gastric cancer patients with *ERCC2* rs13181 and rs1799793, respectively [27, 47], while in East-Asian lung cancer patients, Kim et al. did not find any association between CIN and *ERCC2* rs13181 [48]. *SLC22A2* rs316019, a nonsynonymous missense variant (p.270Ala > Ser; c.808G > T), impacts cisplatin uptake in renal proximal tubules and has been linked to CIN in multiple studies [16,17,21-24]. However, the reported effects of *SLC22A2* rs316019 on CIN vary across populations and outcome definitions. Among East-Asians, Zazuli et al. suggested a protective effect of this SNP

when using CTCAE-AKI criteria [17]. Notably, all these studies used AKI as the endpoint for CIN. In our study, none of the candidate SNPs from *SLC22A2*, *ERCC1* and *ERCC2* showed a significant association with AKD in any genetic model.

Since AKD by itself can lead to poor OS in cancer patients, partly due to the resulting inadequate treatment, we further investigated the effect of these candidate SNPs on OS. In our study, although the OS analysis of AKD vs non-AKD groups was not statistically significant, we observed a trend indicating that patients without AKD had numerically longer OS compared to those with AKD. OS analysis for each selected SNP revealed no association with polymorphisms in *SLC22A2*, *EPHX1* and *ERCC1* genes. However, using a dominant genetic model, OS was associated with *ERCC2* rs13181 and rs1799793: patients with *ERCC2* rs13181-TT genotype and rs1799793-CC genotype exhibited better OS. Notably, these *ERCC2* SNPs showed no association with AKD, suggesting their effects on OS are independent of AKD status. Previously, data on survival in relation to *ERCC2* gene polymorphisms have varied across different populations and cancer types, limiting their application in clinical practice. Variant genotypes of *ERCC2* rs13181 and rs1799793 have been significantly associated with poor survival in Chinese NSCLC patients [49]. A meta-analysis by Yang et al. found that the rs13181 variant G and rs1799793 variant T were poorly associated with OS in NSCLC patients [50]. Among Caucasian gastric cancer patients, rs13181 was not associated with OS, whereas the rs1799793 homozygous variant AA genotype was significantly linked to poor OS [51].

Stratification of OS analysis by cancer type revealed that both SNPs of *ERCC2* were associated only with HNSCC patients in our cohort under a dominant genetic model. Our findings align with recent studies reporting worse OS for HNSCC patients with the *ERCC2* rs13181 homozygous minor genotype and significantly better OS for those with the rs1799793 homozygous major genotype under the same model [52,53]. OS stratification for lung cancer patients showed no significant associations in our study. However, an Asian lung cancer study recently reported that *ERCC2* rs13181 TG and TG+GG genotypes were significantly linked to worse OS under a dominant model [54]. These differing results may stem from the small sample size in our lung cancer cohort. While our findings showed no OS association with *ERCC1* rs11615 and rs3212986, these SNPs have been widely studied in various cancers with varying survival outcomes. The G allele of rs11615 has been significantly associated with better OS, and the A allele of rs3212986 has been linked to improved PFS in a lung cancer study [55]. Conversely, a meta-analysis by Yang et al. found that the *ERCC1* rs11615 T allele was associated with poor OS in NSCLC patients [50].

While this study provides significant insights into the genetic predictors of cisplatin-induced nephrotoxicity, several limitations should be acknowledged. Although we accounted for age and sex in our logistic regression models, other potential confounding factors, such as details of the chemotherapy regimens, pre-chemotherapy hydration status during treatment, and premedication protocols, were not included in the analysis. This is primarily because our study cohort consisted of patients with various tumor types who were treated with different cisplatin-containing regimens, making it challenging to uniformly account for these variables. These factors are known to influence nephrotoxicity outcomes and may have affected our findings. Additionally, the small sample size, single-institution retrospective study design, and limited number of SNPs, all of which are common in Thais, may restrict the generalizability of our findings to broader populations, highlighting the need for validation in larger and more diverse cohorts.

In summary, we demonstrated for the first time that *EPHX1* rs1051740 heterozygous genotype is significantly associated with increased risk of AKD in co-dominant and over-dominant model among Thai cancer patients who received cisplatin. Furthermore, *ERCC2* rs13181 and rs1799793 homozygous major are also confirmed to be associated with better OS in Thai HNSCC patients Table 5. Despite certain limitations, these findings highlight potential prognostic value of the analyzed SNPs, helping clinicians preliminarily identify patients more prone to developing AKD during cisplatin-based treatment, particularly in cases of locally advanced HNSCC, where cisplatin remains a mainstay of concurrent chemoradiotherapy [56]. The AKD-associated SNPs can be utilized to tailor treatment protocols for patients undergoing cisplatin-based therapy, advancing personalized chemotherapy approaches Table 5. Larger prospective studies across diverse populations are necessary to validate these genetic markers for personalized treatment approaches.

**Table 5. Summary of key findings and their clinical implications.**

| Key Findings | Potential clinical implications |
|---|---|
| *EPHX1*<br>rs1051740 | A tailored treatment protocol to reduce potential AKD occurrence (personalized chemotherapy approach) |
| *ERCC2*<br>rs13181<br>rs1799793 | Prognostic biomarkers for HNSCC patients |

## Supporting information

**S1 Fig. Sanger sequencing results for each SNP genotype.**
(TIF)

**S2 Fig. Overall survival analyses of the 121 Head and Neck cancer cohort.**
(TIF)

**S3 Fig. Overall survival analyses of the 39 lung cancer cohort.**
(TIF)

**S4 Fig. Progression-free survival analyses of the 121 Head and Neck cancer cohort.**
(TIF)

**S1 Table. Primer sequences to perform Sanger sequencing.**
(PDF)

**S2 Table. Touchdown PCR conditions for primer optimization.**
(PDF)

**S3 Table. Clinical characteristics of AKI and Non-AKI patients after receiving cisplatin chemotherapy.**
(PDF)

**S4 Table. Association of the selected genetic polymorphisms with cisplatin-induced AKD in Head and Neck cancer cohort.**
(PDF)

**S5 Table. Association of the selected genetic polymorphisms with cisplatin-induced AKD in Head and Neck and Esophageal cancer cohort.**
(PDF)

**S6 Table. Association of the selected genetic polymorphisms with cisplatin-induced AKD in lung cancer cohort.**
(PDF)

**S7 Table. Associated risk factors of Time-to-AKD in this cohort.**
(PDF)

**S8 Table. Risk factors of overall survival.**
(PDF)

**S9 Data File. The genotyping information and clinical metadata required to replicate the reported study findings.**
(XLSX)

## Acknowledgments

We would like to thank Dr. Fah J Sathirapongsasuti for his valuable comments on the data analysis. We are also obliged to Dr. Meng-Shin Shiao and the Ramathibodi research center staff for their input and kind assistance in laboratory work throughout. Our sincere appreciation goes out to the Ramathibodi comprehensive tumor biobank team for their support in retrieval and allocation of patient samples.

## Author contributions

**Conceptualization:** Nuttapong Ngamphaiboon, Natini Jinawath.

**Formal analysis:** Saad Ahmed, Jakris Eu-ahsunthornwattana, Thanaporn Thamrongjirapat, Nintita Sripaiboonkit Thokanit, Nuttapong Ngamphaiboon, Natini Jinawath.

**Funding acquisition:** Natini Jinawath.

**Investigation:** Saad Ahmed, Aruchalean Taweewongsounton, Yutthana Rittavee.

**Resources:** Montien Ngodngamthaweesuk, Pitichote Hiranyatheb, Thanyanan Reungwetwattana, Nuttapong Ngamphaiboon, Natini Jinawath.

**Supervision:** Jakris Eu-ahsunthornwattana, Natini Jinawath.

**Writing – original draft:** Saad Ahmed, Jakris Eu-ahsunthornwattana, Nuttapong Ngamphaiboon, Natini Jinawath.

**Writing – review & editing:** Saad Ahmed, Jakris Eu-ahsunthornwattana, Thanaporn Thamrongjirapat, Aruchalean Taweewongsounton, Yutthana Rittavee, Nintita Sripaiboonkit Thokanit, Montien Ngodngamthaweesuk, Pitichote Hiranyatheb, Thanyanan Reungwetwattana, Nuttapong Ngamphaiboon, Natini Jinawath.

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
