## [Decision Letter · Decision Letter 0]

9 Sep 2024

*EPHX1*  and *ERCC2*

Dear Dr. Jinawath,

Thank you for submitting your manuscript to PLOS ONE. After careful consideration, we feel that it has merit but does not fully meet PLOS ONE’s publication criteria as it currently stands. Therefore, we invite you to submit a revised version of the manuscript that addresses the points raised during the review process.

We look forward to receiving your revised manuscript.

Kind regards,

Emma Campbell, Ph.D

Staff Editor

PLOS ONE

Journal Requirements:

"This research project is supported by Mahidol University, and the National Science Research and Innovation (NSRF) via the Program Management Unit for Human Resources & Institutional Development, Research and Innovation (PMU-B) [Grant number B01F650006]; grant numbers BCG.659 and B05F650041 (NJ)."

Reviewers' comments:

Reviewer's Responses to Questions

**Comments to the Author**

1. Is the manuscript technically sound, and do the data support the conclusions?

Reviewer #1: Yes

Reviewer #2: Yes

Reviewer #3: Yes

2. Has the statistical analysis been performed appropriately and rigorously?

Reviewer #1: N/A

Reviewer #2: Yes

Reviewer #3: Yes

3. Have the authors made all data underlying the findings in their manuscript fully available?

Reviewer #1: Yes

Reviewer #2: No

Reviewer #3: Yes

4. Is the manuscript presented in an intelligible fashion and written in standard English?

Reviewer #1: Yes

Reviewer #2: Yes

Reviewer #3: Yes

Reviewer #1: In their manuscript PONE-D-24-18947 entitled “EPHX1 and ERCC2 polymorphisms are associated with cisplatin induced nephrotoxicity and prognosis in Thai cancer patients”, the authors have demonstrated that Single Nucleotide Polymorphisms (SNPs) can be used to identify cancer patients who are susceptible to developing cisplatin-induced nephrotoxicity (CIN). Their results reveal an association between EPHX1 rs1051740 and AKD, and confirms the previously reported associations between ERCC2 SNPs and OS."

The manuscript is well written, results are clear and the organization of the results makes it easy to follow. The authors have also addressed both strengths and limitations of the study. The article may be suitable for publication in the PLOS ONE journal.

Minor remark: can you provide a short list of abbreviations?

Reviewer #2: The manuscript was well written. Authors presented their findings in a clear and intelligible manner that was easy to follow. Authors may consider shortening the length of the discussion section to ease readability.

A data sharing statement was not included.

Reviewer #3: This manuscript presents significant findings on the genetic predictors of cisplatin-induced nephrotoxicity in Thai cancer patients. It is well-structured, ethically compliant, and methodologically rigorous. The results have considerable potential for clinical application. By identifying genetic markers associated with nephrotoxicity, chemotherapy treatments can be personalized, potentially improving the management of adverse effects. However, there are some limitations that need to be addressed.

Although the study accounts for age and sex in its logistic regression models, other potential confounders such as the details of the chemotherapy regimen, hydration status during treatment, and concurrent medications were not considered. These factors could greatly affect nephrotoxicity outcomes and should be included in the analysis.

The study's sample is restricted to Thai cancer patients treated at a single hospital. This geographic and ethnic uniformity limits the generalizability of the findings to a wider population. Differences in SNP frequencies among various ethnic groups mean that the results may not be applicable to non-Thai populations.

A more detailed discussion of the study’s limitations, including potential biases and the constraints of the analyzed SNPs, would offer a more balanced perspective on the findings and their implications.

**Do you want your identity to be public for this peer review?** For information about this choice, including consent withdrawal, please see our Privacy Policy

Reviewer #1: No

Reviewer #2: No

Reviewer #3: No

---

## [Author Response · Author response to Decision Letter 1]

13 Oct 2024

Response to the requirement #1:

We have revised the formatting according to PLOS ONE’S style requirement.

"This research project is supported by Mahidol University, and the National Science Research and Innovation (NSRF) via the Program Management Unit for Human Resources & Institutional Development, Research and Innovation (PMU-B) [Grant number B01F650006]; grant numbers BCG.659 and B05F650041 (NJ)." Please state what role the funders took in the study. If the funders had no role, please state: "The funders had no role in study design, data collection and analysis, decision to publish, or preparation of the manuscript." If this statement is not correct you must amend it as needed.

Response to the requirement #2:

Please note that the funders had no role in study design, data collection and analysis, decision to publish, or preparation of the manuscript. This statement has been added in both the cover letter and the main manuscript file under the “Funding” heading on page 2. Please see the added text below.

Funding

“This research project is supported by Mahidol University, and the National Science Research and Innovation (NSRF) via the Program Management Unit for Human Resources & Institutional Development, Research and Innovation (PMU-B) [Grant numbers B01F650006, BCG.659 and B05F650041] (NJ). SA received “Master Degree Student Research Assistantship” scholarship from the Faculty of Medicine Ramathibodi hospital, Mahidol University. The funders had no role in study design, data collection and analysis, decision to publish, or preparation of the manuscript.”

Response to the requirement #3:

The corresponding author’s ORCID iD has been updated in the Editorial Manager.

Response to the requirement #4:

We apologize for the overlook of this requirement. Since these data do not form core part of our research, we have revised the information in “Results” section line number 409 and 462 on page 16 and 20, respectively. Please see the revised parts below.

“Upon adjusting the model with other clinical factors such as hypertension, diabetes, heart diseases and cerebrovascular diseases and histological types, also showed TC genotype of EPHX1 rs1051740 to be significantly associated with AKD in both co-dominant and over-dominant models, although the associations were no longer significant after FDR correction“.

“To analyze clinical and genetic factors, multivariate analysis was then performed using the statistically significant variables and potential AKD-associated clinical factors.”

Reviewer #1: In their manuscript PONE-D-24-18947 entitled “EPHX1 and ERCC2 polymorphisms are associated with cisplatin induced nephrotoxicity and prognosis in Thai cancer patients”, the authors have demonstrated that Single Nucleotide Polymorphisms (SNPs) can be used to identify cancer patients who are susceptible to developing cisplatin-induced nephrotoxicity (CIN). Their results reveal an association between EPHX1 rs1051740 and AKD, and confirms the previously reported associations between ERCC2 SNPs and OS." The manuscript is well written, results are clear and the organization of the results makes it easy to follow. The authors have also addressed both strengths and limitations of the study. The article may be suitable for publication in the PLOS ONE journal.

Response to Reviewer #1:

Thank you so much for taking the time to review our manuscript. We do appreciate your constructive comments.

Specific comments:

Minor remark: can you provide a short list of abbreviations?

Response to Reviewer #1:

List of abbreviations used in our manuscript is provided herein and, in the manuscript on page 3.

o CIN: Cisplatin Induced Nephrotoxicity

o AKI: Acute Kidney Injury

o AKD: Acute Kidney Disease

o SNP: Single Nucleotide Polymorphisms

o OCT2: Organic Cation Transporter 2

o SLC22A2: Solute Carrier Family 22 member 2

o ERCC1: Excision Repair Cross-Complementing 1

o ERCC2: Excision Repair Cross-Complementing 2

o EPHX1: Epoxide Hydrolase 1

o XPD: Xeroderma Pigmentosum group D

o DME: Drug Metabolizing Enzyme

o PFS: Progression-Free Survival

o OS: Overall Survival

o CTCAE-AKI: Common Criteria Terminology for Adverse Events - Acute Kidney Injury

o KDIGO: Kidney Disease Improving Global Outcomes

o ICD-10: 10th Revision of International Classification of Diseases

o T-Rex: Thai Reference Exome Database

o MAF: Minor Allele Frequency

o FDR: False Discovery Rate

o BH-FDR: Benjamini-Hochberg False Discovery Rate

o HWE: Hardy-Weinberg Equilibrium

Reviewer #2: The manuscript was well written. Authors presented their findings in a clear and intelligible manner that was easy to follow.

Specific comments:

1. Authors may consider shortening the length of the discussion section to ease readability.

2. A data sharing statement was not included.

Response to Reviewer #2:

Thank you very much for your kind comments and pointing out these points. We have revised the text accordingly as follows.

1. The discussion part has been made more concise where applicable. Please see the revised parts below.

Line 576-580, page 24 Revised: “The use of higher cisplatin doses in the past regimens may be one of the reasons for a higher AKI incidence seen in Patimarattananan et al. The recent improvement of hydration and supplementation regimens may also contribute to lower AKI incidence”.

Line 595 – 599, page 25 Revised: “Similar to our patient’s cohort, male sex, and lower baseline eGFR were among AKD progression factors [35]. Furthermore, a recent meta-analysis identified a higher mortality risk in the AKD group compared to the non-AKD group [34]”.

Line 609 – 615, page 25 Revised: “However, comorbidities including hypertension, diabetes mellitus and age did not show significant association in our study, whereas Patimarattananan et al. suggests that hypertension was significantly associated with AKD development. In line with the same study [1], kidney function after cisplatin treatment did not impact the OS in our study”.

Line 628 – 630, page 26 Deleted: “The extent and outcome of polymorphism’s effect varies, for example, decreased activity of EPHX1 gene is linked to increased risks of having alcohol dependence, liver cirrhosis and breast cancer”.

Line 635 – 638, page 26 Revised: “Lee et al has shown an association between genetically lower epoxide hydrolase activity and an increased risk of tobacco related cancers among smokers”.

Line 656 – 659, page 27 Deleted: “Interestingly, Khrunin et al. who also found rs1051740 heterozygous genotype to be significantly associated with CIN has proposed that EPHX1 might not be acting alone to cause CIN. There may be another gene such as EPHX2 which might exert its effect on the functions of EPHX1, and hence causing CIN”.

Line 711 – 716, page 29 Deleted: “Moreover, studies have shown that both ERCC1 and ERCC2 polymorphisms may influence freedom from loco-regional relapse (ffLRR) and survival outcomes in head & neck and lung cancer patients treated with platinum-based chemotherapies. However, the results from prior studies are inconsistent and does not reach a consensus regarding their clinical use, may be due to sample size and population differences”.

2. Data sharing statement has been included on page 2 of the main manuscript as follows.

Data sharing statement:

“All relevant data are within the manuscript and its supporting information files. We have removed all identifying information. The genotyping information and clinical metadata required to replicate the reported study findings can be found in S7 Data file.”

Reviewer #3: This manuscript presents significant findings on the genetic predictors of cisplatin-induced nephrotoxicity in Thai cancer patients. It is well-structured, ethically compliant, and methodologically rigorous. The results have considerable potential for clinical application. By identifying genetic markers associated with nephrotoxicity, chemotherapy treatments can be personalized, potentially improving the management of adverse effects. However, there are some limitations that need to be addressed.

Specific comments:

1. Although the study accounts for age and sex in its logistic regression models, other potential confounders such as the details of the chemotherapy regimen, hydration status during treatment, and concurrent medications were not considered. These factors could greatly affect nephrotoxicity outcomes and should be included in the analysis.

Response to Reviewer #3:

Thank you very much for the positive feedback and valuable suggestions. We do agree with the reviewer that other potential confounders, such as chemotherapy regimen details, hydration status, and concurrent medications, could influence nephrotoxicity outcomes. However, given that our study included patients with various tumor types treated with different cisplatin-containing regimens, it was challenging to uniformly account for these variables in our analysis. Additionally, detailed data on these factors were not consistently available for all patients, and the complexity of chemotherapy regimens and premedication protocols made it difficult to incorporate them comprehensively. We have acknowledged these limitations in the revised manuscript under the ‘Discussion’ section (Lines 722-729, page 30). Please see the added paragraph below.

“While this study provides significant insights into the genetic predictors of cisplatin-induced nephrotoxicity, several limitations should be acknowledged. Although we accounted for age and sex in our logistic regression models, other potential confounding factors, such as details of the chemotherapy regimens, pre-chemotherapy hydration status during treatment, and premedication protocols, were not included in the analysis. This is primarily because our study cohort consisted of patients with various tumor types who were treated with different cisplatin-containing regimens, making it challenging to uniformly account for these variables. These factors are known to influence nephrotoxicity outcomes and may have affected our findings.”

The study's sample is restricted to Thai cancer patients treated at a single hospital. This geographic and ethnic uniformity limits the generalizability of the findings to a wider population. Differences in SNP frequencies among various ethnic groups mean that the results may not be applicable to non-Thai populations.

2. A more detailed discussion of the study’s limitations, including potential biases and the constraints of the analyzed SNPs, would offer a more balanced perspective on the findings and their implications.

Response to Reviewer #3:

We appreciate your insightfulness and your suggestions regarding the study’s limitation. We have added the potential limitations and biases in more details in the Discussion (Lines 730 -732, page 30). Please see the revised part below.

“Additionally, the small sample size, single-institution retrospective study design, and limited number of SNPs analyzed, all of which are common in Thais may limit the generalizability of our findings to a wider population.”

Additionally, we have revised the last paragraph of the Discussion (Lines 734-743, page 30) to better summarize our findings and their potential application, as shown below:

“In summary, we demonstrated for the first time that EPHX1 rs1051740 heterozygous genotype is significantly associated with increased risk of AKD in co-dominant and over-dominant model among Thai cancer patients who received cisplatin. Furthermore, ERCC2 rs13181 and rs1799793 homozygous major are also confirmed to be associated with better OS in Thai HNSCC patients. Despite certain limitations, these findings highlight potential prognostic value of the analyzed SNPs, helping clinicians preliminarily identify patients more prone to developing AKD during cisplatin-based treatment, particularly in cases of locally advanced HNSCC, where cisplatin remains a mainstay of concurrent chemoradiotherapy (56). Larger prospective studies across diverse populations are necessary to validate these genetic markers for personalized treatment approaches.”

Reference:

56. Ngamphaiboon, N., et al., Evolving role of novel radiosensitizers and immune checkpoint inhibitors in (chemo)radiotherapy of locally advanced head and neck squamous cell carcinoma. Oral Oncol, 2023. 145: p. 106520.

---

## [Decision Letter · Decision Letter 1]

27 Nov 2024

*EPHX1*  and *ERCC2*

Dear Dr. Jinawath,

Thank you for submitting your manuscript to PLOS ONE. After careful consideration, we feel that it has merit but does not fully meet PLOS ONE’s publication criteria as it currently stands. Therefore, we invite you to submit a revised version of the manuscript that addresses the points raised during the review process.

We look forward to receiving your revised manuscript.

Kind regards,

Milad Khorasani, PhD

Academic Editor

PLOS ONE

Reviewers' comments:

Reviewer's Responses to Questions

**Comments to the Author**

Reviewer #4: (No Response)

Reviewer #5: All comments have been addressed

2. Is the manuscript technically sound, and do the data support the conclusions?

Reviewer #4: Yes

Reviewer #5: Yes

3. Has the statistical analysis been performed appropriately and rigorously?

Reviewer #4: No

Reviewer #5: Yes

4. Have the authors made all data underlying the findings in their manuscript fully available?

Reviewer #4: Yes

Reviewer #5: Yes

5. Is the manuscript presented in an intelligible fashion and written in standard English?

Reviewer #4: No

Reviewer #5: Yes

Reviewer #4: The authors report the results of the study “EPHX1 and ERCC2 polymorphisms are associated with cisplatin-induced nephrotoxicity and prognosis in Thai cancer patients”. The authors have evaluated the effects of polymorphisms of six SNPs in the drug-metabolizing enzyme genes, SLC22A2 (rs316019) & EPHX1 (rs1051740), and the DNA repair genes, ERCC1 (rs11615 & rs3212986) & ERCC2 (rs13181 & rs1799793), and CIN in the 169 Thai patients with head and neck, lung, or esophageal cancer and whether they could on predict the cumulative incidence of AKD, progression-free survival (PFS), and overall survival (OS).

While acknowledging the research problem's significance and the contextualisation of results within the existing literature, it is essential to address certain ambiguities in the manuscript. The following significant concerns warrant the authors' attention:

A significant area for improvement of the manuscript is regarding the suitability of the study population to reach the study goal. The sample needs to be more important to achieve the primary objectives. Second, the manuscript is written poorly– which complicates the reading and the precise interpretation of the data.

I have significant comments for this manuscript:

1) The manuscript's language is acceptable but needs to be re-revised; there are word errors on the paper, so it needs to be controlled and corrected.

2) What was the enrollment period? Were all consecutive patients enrolled, and when were the data and follow-up censored?

3) Describe the setting, locations, and relevant dates, including periods of recruitment, exposure, follow-up, and data collection

4) How were the sample size and power calculated for this study?

5) A CONSORT diagram would help better understand which patients were removed from analyses for various reasons.

6) What were the patients' side effects after receiving the platinum chemotherapy? Please elaborate

7) Since chemotherapy is associated with drug toxicity, how dose delay or discontinuance was accounted for?

8) The authors have calculated the odds ratio to calculate the risk. Did the authors adjust the odds with other clinical covariates? Adjusting the odds with different covariates impacts clinical parameters and is crucial.

9) The therapeutic response and survival depend on tumour grade and stage; how was this factor accounted for in the data analysis

10) Cox regression analysis results should be shown (in tables, at least as supplementary data).

11) Another validation cohort should be used to validate the discovery cohort.

12) Insert in the conclusion a sentence or a short sentence that summarises the relevance of the results found in this study. It is essential to emphasise the significance of the results obtained from the objectives proposed by the authors.

13) No attempts have been made to collect information on the other confounding factors which would have influenced for genotyping variants.

14) Some limitations of the study should be discussed.

Reviewer #5: General Comments:

The manuscript, titled "EPHX1 and ERCC2 polymorphisms are associated with cisplatin-induced nephrotoxicity and prognosis in Thai cancer patients", explores an important clinical issue regarding genetic predisposition to cisplatin-induced nephrotoxicity. The study is well-structured and provides significant findings, particularly concerning the association of EPHX1 rs1051740 and ERCC2 SNPs with nephrotoxicity and survival outcomes.

The following comments address the review questions and include additional observations:

Specific Comments on the Manuscript:

Clarity of Hypothesis and Objectives:

The manuscript effectively outlines its objectives and hypotheses, focusing on SNPs linked to nephrotoxicity in Thai cancer patients. However, a clearer differentiation between novel findings and confirmation of existing knowledge would enhance the manuscript's impact.

Data Availability and Sharing:

While the authors have addressed data sharing requirements, ensuring that the data and supplementary files (e.g., S7 Data file) are easily accessible is crucial. Clarification regarding the repository details would improve transparency.

Methods:

The methods are detailed and appropriate for the study's aims. However, the choice of SNPs and their allele frequencies should be briefly discussed in the context of Thai population genetics.

Missing details on hydration regimens, concurrent medications, and cisplatin dosing specifics should be acknowledged as potential confounders.

Results:

The results are systematically presented, but some sections could benefit from additional context:

Highlight the clinical relevance of the significant findings, especially the EPHX1 rs1051740 association with AKD.

The Kaplan-Meier curves and survival analyses are insightful, though more detailed figures or supplementary visuals might aid interpretation.

Discussion:

While the discussion is comprehensive, consider further condensing to maintain focus on the key findings.

Address the implications of the findings for clinical practice, particularly regarding personalized chemotherapy approaches.

Study Limitations:

The authors acknowledge the study's limitations, including geographic and ethnic constraints. Emphasizing the need for validation in diverse cohorts would strengthen the discussion.

Ethical and Publication Concerns:

Ethical Compliance:

The manuscript states approval from the Human Research Ethics Committee and adherence to the Declaration of Helsinki. This is sufficient, though explicit mention of informed consent obtained for SNP analysis would reinforce compliance.

Publication Ethics:

There are no indications of dual publication or ethical concerns.

Recommendations for Improvement:

Include a concise abbreviation list in the manuscript's main body to improve readability.

Ensure that all cited data is either included in the manuscript or accessible through provided links.

Provide a summary table of key findings with clinical implications to improve accessibility for non-specialist readers.

Final Evaluation:

The manuscript is well-written and presents valuable findings that contribute to the field of oncology and pharmacogenomics. After addressing the aforementioned comments, the manuscript will be suitable for publication in PLOS ONE.

**Do you want your identity to be public for this peer review?** For information about this choice, including consent withdrawal, please see our Privacy Policy

Reviewer #4: No

Reviewer #5: **Yes: ** Kazuo Kobayashi

---

## [Author Response · Author response to Decision Letter 2]

1 Jan 2025

Reviewer’s Comments:

Reviewer #4: The authors report the results of the study “EPHX1 and ERCC2 polymorphisms are associated with cisplatin-induced nephrotoxicity and prognosis in Thai cancer patients”. The authors have evaluated the effects of polymorphisms of six SNPs in the drug-metabolizing enzyme genes, SLC22A2 (rs316019) & EPHX1 (rs1051740), and the DNA repair genes, ERCC1 (rs11615 & rs3212986) & ERCC2 (rs13181 & rs1799793), and CIN in the 169 Thai patients with head and neck, lung, or esophageal cancer and whether they could on predict the cumulative incidence of AKD, progression-free survival (PFS), and overall survival (OS).

While acknowledging the research problem's significance and the contextualisation of results within the existing literature, it is essential to address certain ambiguities in the manuscript. The following significant concerns warrant the authors' attention:

A significant area for improvement of the manuscript is regarding the suitability of the study population to reach the study goal. The sample needs to be more important to achieve the primary objectives. Second, the manuscript is written poorly– which complicates the reading and the precise interpretation of the data.

I have significant comments for this manuscript:

1) The manuscript's language is acceptable but needs to be re-revised; there are word errors on the paper, so it needs to be controlled and corrected.

Response to the Reviewer # 4:

Thank you for your kind suggestion. We have thoroughly checked the grammar and corrected the writing throughout the manuscript, where applicable:

Page 5 – Lines 126-130: “Despite being an effective anticancer drug, its usage can be delayed or terminated because of associated toxicities such as neurotoxicity, gastrointestinal toxicity, ototoxicity and nephrotoxicity [1]. Platinum could be retained in the plasma for as long as 20 years after discontinuation of cisplatin-based chemotherapy [2, 3], thus increasing concerns about the long-term nephrotoxicity risks [4].”

Pages 5-6 – Lines 143-152: “Clinically, CIN manifests in various forms, with Acute Kidney Injury (AKI) being the most common, occurring in 20-30% of patients [7]. Animal models of CIN demonstrates increased serum creatinine (SCr) and blood urea nitrogen, along with histopathological changes [8]. Though SCr is the main marker for renal function evaluation, it needs to be considered in conjunction with other factors such as age and sex. CIN is dose-dependent, with lower doses of cisplatin causing delayed kidney damage as compared to higher doses [9]. Cisplatin dosing regimens vary by cancer type, leading to differences in nephrotoxicity incidence. In lung and esophageal cancer treatments, ≥ 60 mg/m2 every 3 - 4 weeks and 75 mg/m2 on days 1 and 29 are administered, respectively [10, 11]. Gradually, in HNSCC, weekly 40 mg/m2 or 100mg/m2 every 3 weeks is being administered concurrently with radiotherapy [12].”

Page 6 – Line 154-159: “AKI may occur within 7 days of cisplatin administration, presented as increase in SCr, whereas Acute Kidney Disease (AKD) develops between 7 and 90 days of cisplatin initiation [13]. Prior studies have commonly used AKI as an endpoint for CIN based on different grading criteria; while the long-term or delayed effects of CIN are not widely studied. AKD, defined as renal damage occurring within 3 months of cisplatin administration, may prove to be a valuable endpoint for studying its delayed nephrotoxic effects [1, 14].”

Page 7 – Lines 178-185: “ERCC1 and ERCC2 or XPD are integral part of the NER pathway. SNPs in these genes - ERCC1 rs11615 and rs3212986, and ERCC2 rs13181 and rs1799793 - are linked to changes in the DNA repair process. Variations in these genes may cause a decrease in the helicase activity and DNA repair. ERCC1 plays a role in incising the DNA damage site, and can cause a rate-limiting effect, while ERCC2, is involved in helicase unwinding the DNA at the damaged site. Hence, these SNPs may reduce the efficiency of DNA repair mechanisms, leading to greater platinum-DNA adduct formation and hindered nephron repair following cisplatin exposure [19].”

Page 25 – Lines 576-580: “We found EPHX1 rs1051740 significantly associated with cisplatin-induced AKD in co-dominant and over-dominant model, particularly in TC pattern. To our knowledge, no prior studies have performed AKD association analysis with genetic polymorphisms. Multivariate analysis, identified lower baseline eGFR less than 90 ml/min/1.73m2 as another factor influencing AKD development after cisplatin treatment.”

Page 25 – Lines 586-596: “No consensus has been reached regarding SNPs associated with CIN, partly due to significant variations in minor allele frequencies across populations. Previous studies have shown association of these 6 selected SNPS with CIN using AKI as the endpoint. Notably, the minor allele frequency of EPHX1 rs1051740 differs in the Thai population compared to other populations [36]. Most commonly the reference allele is T, whose expression as homozygous TT genotype is associated with faster metabolism than CC. However, in Thai population, T is the minor allele, with a frequency of 0.49. The EPHX1 rs1051740 (c.337T>C), is one of the 2 widely studied polymorphisms of the EPHX1 gene. This variant results in a tyrosine-to- histidine substitution at position 113, leading to a 50% reduction in enzyme activity [37]. Decreased enzymatic activity may reduce the detoxification, increasing the formation of highly reactive metabolites, making the cells more susceptible to challenges.”

Page 26 – Lines 603-607: “Another meta-analysis found an association between rs1051740 polymorphism and HNSCC risk in population-based studies. [18]. While EPHX1 is well-studied regarding cancer risk, few studies have explored its relationship with CIN. To date, only Khrunin et al. have reported a higher frequency of CIN in ovarian cancer patients with the rs1051740 heterozygous genotype in a Russian population [15].”

Pages 26-27 – Lines 621-640: “ERCC1 and ERCC2 are key components of the NER pathway, activated by the formation of DNA adducts. Genetic variations in these genes may influence DNA repair capacity. Synonymous variants in the 3’ UTR region such as ERCC1 rs11615 and rs3212986, can affect protein stability, folding, structure, and expression [19, 45]. Previous studies by Khrunin et al. and Tzvetkov et al. reported an increased risk of CIN in ovarian and other cancers related to ERCC1 rs11615 and rs3212986 in Caucasian [25, 26]. However, Zazuli et al. found ERCC1 rs3212986 to reduce risk of CIN in Caucasian testicular cancer patients [17]. In East-Asians, Chen et al. found no significant association between CIN and ERCC1 rs11615 in Chinese lung cancer patients [46]. The association of ERCC2 with CIN has also been widely studied, but results vary due to differing endpoints and populations. Studies have shown an increased risk of CIN in Caucasian osteosarcoma and gastric cancer patients with ERCC2 rs13181 and rs1799793, respectively [27, 47], while in East-Asian lung cancer patients, Kim et al. did not find any association between CIN and ERCC2 rs13181 [48]. SLC22A2 rs316019, a nonsynonymous missense variant (p.270Ala>Ser; c.808G>T), impacts cisplatin uptake in renal proximal tubules and has been linked to CIN in multiple studies [16, 17, 21-24]. However, the reported effects of SLC22A2 rs316019 on CIN vary across populations and outcome definitions. Among East-Asians, Zazuli et al. suggested a protective effect of this SNP when using CTCAE-AKI criteria [17]. Notably, all these studies used AKI as the endpoint for CIN. In our study, none of the candidate SNPs from SLC22A2, ERCC1 and ERCC2 showed a significant association with AKD in any genetic model.”

Pages 27-28 – Lines 642-657: “Since AKD by itself can lead to poor OS in cancer patients, partly due to the resulting inadequate treatment, we further investigated the effect of these candidate SNPs on OS. In our study, although the OS analysis of AKD vs non-AKD groups was not statistically significant, we observed a trend indicating that patients without AKD had numerically longer OS compared to those with AKD. OS analysis for each selected SNP revealed no association with polymorphisms in SLC22A2, EPHX1 and ERCC1 genes. However, using a dominant genetic model, OS was associated with ERCC2 rs13181 and rs1799793: patients with ERCC2 rs13181-TT genotype and rs1799793-CC genotype exhibited better OS. Notably, these ERCC2 SNPs showed no association with AKD, suggesting their effects on OS are independent of AKD status. Previously, data on survival in relation to ERCC2 gene polymorphisms have varied across different populations and cancer types, limiting their application in clinical practice. Variant genotypes of ERCC2 rs13181 and rs1799793 have been significantly associated with poor survival in Chinese NSCLC patients [49]. A meta-analysis by Yang et al. found that the rs13181 variant G and rs1799793 variant T were poorly associated with OS in NSCLC patients [50]. Among Caucasian gastric cancer patients, rs13181 was not associated with OS, whereas the rs1799793 homozygous variant AA genotype was significantly linked to poor OS [51].”

Page 28 – Lines 659-670: “Stratification of OS analysis by cancer type revealed that both SNPs of ERCC2 were associated only with HNSCC patients in our cohort under a dominant genetic model. Our findings align with recent studies reporting worse OS for HNSCC patients with the ERCC2 rs13181 homozygous minor genotype and significantly better OS for those with the rs1799793 homozygous major genotype under the same model [52, 53]. OS stratification for lung cancer patients showed no significant associations in our study. However, an Asian lung cancer study recently reported that ERCC2 rs13181 TG and TG+GG genotypes were significantly linked to worse OS under a dominant model [54]. These differing results may stem from the small sample size in our lung cancer cohort. While our findings showed no OS association with ERCC1 rs11615 and rs3212986, these SNPs have been widely studied in various cancers with varying survival outcomes. The G allele of rs11615has been significantly associated with better OS, and the A allele of rs3212986 has been linked to improved PFS in a lung cancer study [55].”

Reviewer#4:

2) What was the enrollment period? Were all consecutive patients enrolled, and when were the data and follow-up censored?

Response to the Reviewer # 4:

Thank you for giving us the opportunity to elaborate the enrollment period further, please see the revised parts in the manuscript and as follows:

Pages 8-9 – Lines 218-223: “For HNSCC cohort, in our prospective multidisciplinary observation study of HNSCC and NPC patients at Ramathibodi Hospital, Mahidol University, we have accrued eligible patients since 2016. The study enrolled patients with LA-HNSCC and NPC treated according to the standard of care at their treating physician’s discretion. Enrollment was consistent, except during Covid-19 pandemic when the enrollment was hold, and subsequently resumed in 2021. Similarly, lung and esophageal cancer cohorts were also enrolled since 2016.”

Page 11 – Lines 300-301: “The survival status was cross-checked with the National Security Death Index of Thailand on 23/03/2022.”

Reviewer#4:

3) Describe the setting, locations, and relevant dates, including periods of recruitment, exposure, follow-up, and data collection

Response to the Reviewer # 4:

We appreciate the reviewer for their insightful comment. All enrolled patients were treated with cisplatin at Faculty of Medicine, Ramathibodi Hospital, Mahidol University, Bangkok, Thailand and were retrospectively identified with their available blood samples in Ramathibodi Comprehensive Tumor Biobank collected between 1st January 2016 to 30th June 2020. We also described the details in the “Patient Selection” section on Pages 8-9 – Lines 200-223 and here as follows:

“Clinical and treatment data were obtained from an in-house observational head and neck cancer cohort database, Ramathibodi Tumor Registry, pharmacy database, and medical records included patient demographics, serum creatinine (SCr), estimated glomerular filtration rate (eGFR; CKD-EPI), cancer staging and comorbidities (ICD-10). All clinical information was verified by two independent medical oncologists. The clinical data were initially accessed on 15th July 2020. The authors had access to information that could identify individual participants during data collection; however, no individually identifiable data was reported in this study.

Solid tumor patients treated with cisplatin chemotherapy at the Faculty of Medicine, Ramathibodi Hospital, Mahidol University, Bangkok, Thailand were retrospectively identified. Criteria used for patient inclusion were: head & neck, lung and esophageal cancer patients, age ≥18 years, availability of blood samples in Ramathibodi Comprehensive Tumor Biobank collected between 1st January 2016 to 30th June 2020. Samples were collected from the biobank for research purposes starting from 17th March 2021. Patients with known chronic kidney disease as defined by The Kidney Disease Improving Global Outcomes (KDIGO) guideline before the start of cisplatin-based treatment were excluded [14].

For HNSCC cohort, in our prospective multidisciplinary observation study of HNSCC and NPC patients at Ramathibodi Hospital, Mahidol University, we have accrued eligible patients since 2016. The study enrolled patients with LA-HNSCC and NPC treated according to the standard of care at their treating physician’s discretion. Enrollment was consistent, except during Covid-19 pandemic when the enrollment was hold, and subsequently resumed in 2021. Similarly, lung and esophageal cancer cohorts were also enrolled since 2016. The study enrolled patients with LA-HNSCC and NPC treated according to the standard of care at their treating physician’s discretion. Enrollment was consistent, except during Covid-19 pandemic when the enrollment was hold, and subsequently resumed in 2021. Similarly, lung and esophageal cancer cohorts were also enrolled since 2016.”

Reviewer#4:

4) How were the sample size and power calculated for this study?

Response to the Reviewer # 4:

Thank you for your comment. In our study, sample size and power were calculated using the STATA program, assuming 25% nephrotoxicity, which indicated 80% power in 169 patients. This allowed us to identify a statistically significant association between selected SNPs and AKD.

We have also mentioned power and sample size calculation on Page 11 – Lines 282 - 284 and here as follows:

“Power calculation showed that 169 patients assuming 25% nephrotoxicity would give our study 80% power to identify statistically significant association between selected SNPs and AKD with an alpha of 0.05 and odds ratio of 2.8.”

Reviewer#4:

5) A CONSORT diagram would help better understand which patients were removed from analyses for various reasons.

Response to the Reviewer # 4:

Thank you for your kind suggestion, we have added a CONSORT diagram as Figure 1 in the main figures and as follows:

Reviewer#4:

6) What were the patients' side effects after receiving the platinum chemotherapy? Please elaborate

Response to the Reviewer # 4:

We sincerely thank the reviewer for their insightful comment regarding the side effects experienced by patients after receiving platinum chemotherapy. In this study, we specifically focused on SNPs associated with cisplatin-induced nephrotoxicity. Consequently, the only toxicity data collected and analyzed pertained to nephrotoxicity. Unfortunately, information on other toxicities was not available in this dataset.

We acknowledge the importance of capturing a broader spectrum of chemotherapy-related toxicities and will consider this aspect for future research to provide a more comprehensive analysis.

Reviewer#4:

7) Since chemotherapy is associated with drug toxicity, how dose delay or discontinuance was accounted for?

Response to the Reviewer # 4:

Thank you for your valuable comment. Consistently accounting for this potential confounder was challenging due to the varying dosing regimens across tumor types. We acknowledge it’s imp

---

## [Decision Letter · Decision Letter 2]

11 Mar 2025

*EPHX1*  and *ERCC2*

Dear Dr. Jinawath,

Thank you for submitting your manuscript to PLOS ONE. After careful consideration, we feel that it has merit but does not fully meet PLOS ONE’s publication criteria as it currently stands. Therefore, we invite you to submit a revised version of the manuscript that addresses the points raised during the review process.

We look forward to receiving your revised manuscript.

Kind regards,

Milad Khorasani, PhD

Academic Editor

PLOS ONE

Journal Requirements:

Reviewers' comments:

Reviewer's Responses to Questions

**Comments to the Author**

Reviewer #5: All comments have been addressed

Reviewer #6: All comments have been addressed

2. Is the manuscript technically sound, and do the data support the conclusions?

Reviewer #5: Yes

Reviewer #6: Yes

3. Has the statistical analysis been performed appropriately and rigorously?

Reviewer #5: Yes

Reviewer #6: Yes

4. Have the authors made all data underlying the findings in their manuscript fully available?

Reviewer #5: Yes

Reviewer #6: Yes

5. Is the manuscript presented in an intelligible fashion and written in standard English?

Reviewer #5: Yes

Reviewer #6: Yes

Reviewer #5: The manuscript, titled "EPHX1 and ERCC2 polymorphisms are associated with cisplatin-induced nephrotoxicity and prognosis in Thai cancer patients," addresses a clinically relevant issue and presents robust findings on the association of genetic polymorphisms with cisplatin-induced nephrotoxicity and survival outcomes. The study is well-structured, with clear objectives, detailed methods, and comprehensive results.

Strengths:

The study investigates a significant clinical problem and fills gaps in existing knowledge regarding the association of SNPs with nephrotoxicity and survival in Thai cancer patients.

The inclusion of SNPs relevant to drug metabolism and DNA repair pathways enhances the study's translational potential.

The manuscript demonstrates clear implications for personalized chemotherapy, particularly in tailoring cisplatin-based treatments.

Recommendations:

The authors have adequately addressed the initial concerns and incorporated suggestions to improve the manuscript's clarity, focus, and presentation.

The addition of a CONSORT diagram and a key findings table significantly enhances the manuscript's accessibility for readers.

The discussion of limitations is transparent, and the authors have acknowledged the need for validation in larger, diverse cohorts.

Conclusion:

The manuscript is scientifically sound, addresses all reviewer concerns comprehensively, and meets the journal's standards for publication. It provides valuable insights that are likely to advance personalized medicine in oncology.

I recommend accepting the manuscript for publication in its current form.

Reviewer #6: Your study makes a significant contribution to understanding the genetic factors associated with AKD and demonstrates a well-thought-out approach to data analysis. Overall, your study is valuable and deserves attention within the research community. However, certain aspects, such as the choice of correction methods and inheritance models, could be further discussed to refine the conclusions.

1. Table 2 (L404) presents a comparison of allele frequencies between the T-REx database and the current study (based on the frequencies observed in non-AKD patients). Additionally, for better comparison, global MAF frequencies should be included in the table.

2. False Discovery Rate (FDR) was used for multiple comparisons, which is commonly applied in large-scale studies such as GWAS. However, given the small sample size and a limited number of comparisons (169 samples and 6 SNPs), Bonferroni correction is preferable to minimize false positives. In this case, a more balanced approach could be the Holm correction, which is less conservative than Bonferroni but provides better Type I error control than FDR.

3. Since the researchers applied five genetic models (codominant, dominant, recessive, overdominant, and additive), power analysis was conducted for each of them. However, the additive model is the most justified approach, as it is widely used in SNP association studies due to its general applicability and suitability for small sample sizes. Given the study parameters (n = 169, 6 SNPs), the additive model remains the optimal choice for power calculations, as it accounts for all possible genotypes and provides better sensitivity. However, if the minor allele frequency is particularly low, the dominant model should also be considered.

4. L482 (R2): The TC genotype of EPHX1 rs1051740 was significantly associated with an increased risk of AKD in the codominant model [OR 13.333, 95% CI 1.684 - 105.533, P = 0.014].

The confidence interval (CI) of 1.684 - 105.533 is extremely wide, indicating high uncertainty (low precision) in the effect estimation. This result cannot be considered reliable. Its interpretation should be approached with caution, as the large uncertainty makes the odds ratio (OR) estimate questionable. It is essential to check for the potential influence of outliers and conduct bootstrap estimation of the CI to assess its stability.

6.Were other factors, such as hydration status, concurrent medications, and chemotherapy regimens, considered in the study? Their incomplete accounting could introduce bias into the results.

**Do you want your identity to be public for this peer review?** For information about this choice, including consent withdrawal, please see our Privacy Policy

Reviewer #5: **Yes: ** Kazuo Kobayashi

Reviewer #6: **Yes: ** Alisher Abdullaev

---

## [Author Response · Author response to Decision Letter 3]

27 Mar 2025

Reviewer’s Comments:

Reviewer #6:

1) Table 2 (L404) presents a comparison of allele frequencies between the T-REx database and the current study (based on the frequencies observed in non-AKD patients). Additionally, for better comparison, global MAF frequencies should be included in the table.

Response to the Reviewer # 6:

Thank you for your valuable suggestion. Adding global MAF would certainly provide a better comparison. we have added the revised Table 2 on Page 16 and here:

Gene SNP SNP Information Global Frequencies T-REx Frequencies Data Allele Frequencies HWE P value

SLC22A2 rs316019 Missense c.808G>T (p.270Ala > Ser) C A C A C A 0.187

0.895 0.104 0.890 0.11 0.8289 0.1711

EPXH1 rs1051740 Missense c.337T>C (p.Tyr113His) C T C T C T 0.331

0.295 0.704 0.504 0.496 0.5036 0.4964

ERCC1 rs11615 Synonymous c.354T>C (p.Asn118Asn) G A G A G A 0.482

0.414 0.585 0.697 0.303 0.6 0.4

ERCC1 rs3212986 Synonymous 3’ UTR c*197G>T C A C A C A 0.261

0.745 0.254 0.664 0.336 0.7205 0.2795

ERCC2 rs13181 Missense c.2251A>C (p.751lys > Gln) T G T G T G 0.91

0.642 0.000 0.901 0.099 0.9108 0.0892

ERCC2 rs1799793 Missense c.934G>A (p.Asp312Asn) C T C T C T 0.889

0.684 0.315 0.937 0.063 0.9422 0.0578

Global frequencies were obtained from dbSNP (build 157), accessed on 13th March 2025.

The following sentences “Allele frequencies in Thai and global populations were respectively obtained from the Thai Reference Exome Database (T-REx), which contains data from 1,092 Thai individuals (accessed on 25th February 2022), and from dbSNP (build 157) (accessed on 13th March 2025) & “Allele frequencies were obtained from dbSNP (build 157; https://www.ncbi.nlm.nih.gov/snp/), and from the T-REx database [30]. are also added in the Materials and Methods (Line 248-251, pages 9-10) and table legend (Line 405-406, page 16), respectively.

2) False Discovery Rate (FDR) was used for multiple comparisons, which is commonly applied in large-scale studies such as GWAS. However, given the small sample size and a limited number of comparisons (169 samples and 6 SNPs), Bonferroni correction is preferable to minimize false positives. In this case, a more balanced approach could be the Holm correction, which is less conservative than Bonferroni but provides better Type I error control than FDR.

Response to the Reviewer # 6:

Thank you for your insightful comment. We have performed the Holm-Bonferroni correction for our main results and revised Table 3 accordingly in the manuscript and here as follows:

These sentences “Multiple comparisons were corrected by Benjamini-Hochberg false discovery rate (FDR) corrected P-values, and additionally with Holm-Bonferroni correction” & “although the associations were no longer significant after FDR and Holm-Bonferroni corrections” were also added in the Materials and Methods (Line 296-297, page 11) and Results (Line 421-422, page 17), respectively.

3) Since the researchers applied five genetic models (codominant, dominant, recessive, overdominant, and additive), power analysis was conducted for each of them. However, the additive model is the most justified approach, as it is widely used in SNP association studies due to its general applicability and suitability for small sample sizes. Given the study parameters (n = 169, 6 SNPs), the additive model remains the optimal choice for power calculations, as it accounts for all possible genotypes and provides better sensitivity. However, if the minor allele frequency is particularly low, the dominant model should also be considered.

Response to the Reviewer # 6

Thank you for your suggestion. Previously, we have calculated the power for the overall study regardless of the genetic model (Lines 283 - 286, Page 11). As recommended, we have performed power calculation for additive and dominant model. Details of genetic model power calculation were added in the manuscript (Lines 291 - 294, Page 11) and here. Results varied across the two models, 53% being the highest. While it is below the conventional threshold of 80%, suggesting limited statistical power to detect significant association due to small size, we acknowledged this as a study limitation in the manuscript (Lines 686 - 689, Page 29) and here.

Lines 283 - 286, Page 11: “Power calculations showed that a sample size of 169 patients, assuming 25% nephrotoxicity, would provide our study with 80% power to identify a statistically significant association between the selected SNPs and AKD, with an alpha of 0.05 and an odds ratio of 2.8.

Lines 291 - 294, Page 11 “For the additive and dominant genetic models, power calculations were conducted using STATA 14, power two proportions and RStudio 4.4.2 genpwr package, respectively. For the six SNPs analyzed, the estimated power varied with 53% being the highest”.

Lines 686 - 689, Page 29: “Additionally, the small sample size, single-institution retrospective study design, and limited number of SNPs, all of which are common in Thais, may restrict the generalizability of our findings to broader populations, highlighting the need for validation in larger and more diverse cohorts”.

4) L482 (R2): The TC genotype of EPHX1 rs1051740 was significantly associated with an increased risk of AKD in the codominant model [OR 13.333, 95% CI 1.684 - 105.533, P = 0.014].

The confidence interval (CI) of 1.684 - 105.533 is extremely wide, indicating high uncertainty (low precision) in the effect estimation. This result cannot be considered reliable. Its interpretation should be approached with caution, as the large uncertainty makes the odds ratio (OR) estimate questionable. It is essential to check for the potential influence of outliers and conduct bootstrap estimation of the CI to assess its stability.

Response to the Reviewer # 6

Table 4.1: Genotype frequency of EPHX1 rs1051740 under the codominant model in the HNSCC-only cohort

Codominant model No AKD AKD Total

CC 26 (96.3) 1 (3.7) 27 (100)

TC 39 (66.1) 20 (33.9) 59 (100)

TT 29 (82.86) 6 (17.14) 35 (100)

Total 94 (77.69) 27 (22.31) 121 (100)

We appreciate your observations regarding wide confidence interval (CI) of SNP rs1051740 and the reliability of this result.

Comparing to the combined three-cancer-type cohort, the sample size of our head and neck cancer-only cohort is smaller, and only has one case with the “CC” EPHX1 rs1051740 genotype (Table 4.1). This by itself could lead to the overestimation of results and contribute to a wider CI. To address this concern, we performed a bootstrap analysis with 1,000 repetitions. The bootstrap estimates, including the odds ratio (OR) and CI, are now included in the updated Supplementary Table S4A. Also, we added the sentences here: “Of note, bootstrap estimation was performed for the head and neck cancer-only cohort to correct for the wide confidence interval, which could lead to overestimation of the results S4A Table”. & “Our association analysis results show EPHX1 rs1051740 heterozygous genotype to be associated with AKD in our combined three-cancer-type cohort. This SNP was also significantly associated with AKD in the head and neck cancer-only cohort, although the confidence interval (CI) was wide, likely due to the smaller sample size” in our Results (Lines 489-491, Page 22) & Discussion (Lines 611 - 614, Page 26), respectively.

Additionally, we conducted a comparison between 1,000 and 10,000 bootstrap repetitions for this specific SNP. We found that the effect size (OR) remained consistent across both bootstrap iterations, and the improved confidence intervals were comparable when both repetitions were

used (Table 4.2). Based on these findings, we opted to use 1,000 bootstrap repetitions for all other SNPs and models as shown in the revised Supplementary Table S4A.

Furthermore, the model demonstrated good reliability and validity, with an AUC of 0.77 (Figure 4.1) and a non-significant Hosmer-Lemeshow test (p = 0.571) (Table 4.3) after adjusting for age and sex. These results suggest a good model fit and support the robustness of our findings.

Table 4.2: Bootstrap estimation comparison between 1,000 and 10,000 repetitions in EPHX1 rs1051740 co-dominant model in HNSCC-only cohort.

Bootstrap estimation repetitions

Co-dominant AKD Non-AKD 1,000 repetitions P value 10,000 repetitions P value

CC 1 (3.7) 26 (27.7) 1 1

TC 20 (74.1) 39 (41.5) 17.319 (5.131-58.460) <0.001 17.319 (5.131-58.460) <0.001

TT 6 (22.2) 29 (30.8) 5.824 (1.357-24.985) 0.018 5.824 (1.408 - 24.080) 0.015

Figure 4.1: AUC for the EPHX1 rs1051740

Table 4.3: Hosmer-Lemeshow test for EPHX1 rs1051740 in HNSCC-only cohort shows a non-significant result

Logistic model for AKD event goodness-of-fit test:

Number of observations 121

Number of covariate patterns 119

Pearson chi2(114) 110.64

Prob > chi2 0.5715

5) Were other factors, such as hydration status, concurrent medications, and chemotherapy regimens, considered in the study? Their incomplete accounting could introduce bias into the results.

Response to the Reviewer # 6:

Thank you for raising this valid concern. We agree that these factors could introduce bias into the results. However, we acknowledge these as limitations of our study and addressed this in the manuscript on Line 678 – 685, Page 29 and here as follows:

“While this study provides significant insights into the genetic predictors of cisplatin-induced nephrotoxicity, several limitations should be acknowledged. Although we accounted for age and sex in our logistic regression models, other potential confounding factors, such as details of the chemotherapy regimens, pre-chemotherapy hydration status during treatment, and premedication protocols, were not included in the analysis. This is primarily because our study cohort consisted of patients with various tumor types who were treated with different cisplatin-containing regimens, making it challenging to uniformly account for these variables. These factors are known to influence nephrotoxicity outcomes and may have affected our findings. “

---

## [Editor Report · Decision Letter 3]

30 Apr 2025

*EPHX1*  and *ERCC2* polymorphisms are associated with cisplatin induced nephrotoxicity and prognosis in Thai cancer patients

PONE-D-24-18947R3

Dear Dr. Jinawath,

We’re pleased to inform you that your manuscript has been judged scientifically suitable for publication and will be formally accepted for publication once it meets all outstanding technical requirements.

Kind regards,

Milad Khorasani, PhD

Academic Editor

PLOS ONE
---

## [Editor Report · Acceptance letter]

PONE-D-24-18947R3

PLOS ONE

Dear Dr. Jinawath,

I'm pleased to inform you that your manuscript has been deemed suitable for publication in PLOS ONE. Congratulations! Your manuscript is now being handed over to our production team.

Kind regards,

on behalf of

Dr. Milad Khorasani

Academic Editor

PLOS ONE